# TARSS-Net: Temporal-Aware Radar Semantic Segmentation Network

**Youcheng Zhang**[1][*]  **Liwen Zhang**[1][*][†]  **Zijun Hu**[1]  **Pengcheng Pi**[1]
**Teng Li**[2]  **Yuanpei Chen**[1]  **Shi Peng**[1]  **Zhe Ma**[1][†]
[1]Intelligent Science and Technology Academy of CASIC
[2]Shenzhen International Graduate School, Tsinghua University
youcheng17@163.com[*] lwzhang9161@126.com[*][†] mazhe_thu@163.com[†]

## Abstract

Radar signal interpretation plays a crucial role in remote detection and ranging. With the gradual display of the advantages of neural network technology in signal processing, learning-based radar signal interpretation is becoming a research hot-spot and made great progress. And since radar semantic segmentation (RSS) can provide more fine-grained target information, it has become a more concerned direction in this field. However, the temporal information, which is an important clue for analyzing radar data, has not been exploited sufficiently in present RSS frameworks. In this work, we propose a novel temporal information learning paradigm, *i.e.*, **data-driven temporal information aggregation with learned target-history relations**. Following this idea, a flexible learning module, called **T**emporal **R**elation-**A**ware **M**odule (TRAM) is carefully designed. TRAM contains two main blocks: i) an encoder for capturing the target-history temporal relations (TH-TRE) and ii) a learnable temporal relation attentive pooling (TRAP) for aggregating temporal information. Based on TRAM, an end-to-end **T**emporal-**A**ware **RSS Net**work (TARSS-Net) is presented, which has outstanding performance on publicly available and our collected real-measured datasets. Code and supplementary materials are available at https://github.com/zlw9161/TARSS-Net.

## 1 Introduction

Radar is a reliable remote sensing device for its robustness in adverse weather and illumination conditions. It is widely used for applications like the autonomous driving [33, 21, 36], Unmanned Aerial Vehicle (UAV) surveillance [9, 19], sea monitoring [31, 24, 32], etc. However, the received scattered radar signals are high-entropy information body coupled with environmental clutters from a large spatial range, device noise and moving target information (*i.e.*, distance/range, direction/angle and velocity/Doppler shift). These components are additionally condensed into the received echoes, which is inherently a gap in understanding of radar signals to human perception, as opposed to visible light images, which are modalities aligned with human visual perception, and natural language texts that are naturally based on human semantic understanding. This human-unfriendly perception gap causes great difficulty in semantic perception of scenes and objects [22]. More information on radar signals and processing is in Appendix A.

Encouraged by the success of deep learning techniques in computer vision, in especial, object detection [11, 17, 27, 26] and semantic segmentation [20, 28, 2, 10], some efforts have been made recently to better understand complex radar data. Compared with detection models [33, 6] using bounding boxes, the segmentation models [1, 14, 21, 36, 35] can provide *pixel-wise* (a unit of the

---

[*]Equal contributions.

[†]Corresponding author.

range-angle (RA) or range-doppler (RD) frequency representation) detection results for objects and even the background, which is practical for radar scene understanding and is also followed by this work. Most of these methods are based on convolutional auto-encoding-decoding (CAED), which take the frequency representations of a series of fast Fourier transforms (FFTs) on the radar signals as input, and make predictions on the RA or RD view or both views.

By analysing current efforts in RSS, two primarily concerned problems can be concluded for effective design of RSS system. i) *How to capture target signature effectively in spatial domain*. Typically, atrous spatial pyramid pooling (ASPP) [3] is used to obtain multi-scale spatial information [14, 21], and deformable convolution is utilized to extract target features with irregular shape [6]. Specifically, PeakConv (PKC) [35] is the learning-based operator tailored for radar signals which aims to capture target signature's salience. ii) *How to utilize inherent temporal information of radar input tensor*. Commonly, 3D convolution (3DConv) is implemented [21, 33, 6]. It is worth noting that, compared with the direct use of existing methods, taking into account the intrinsic merits of radar signal can achieve more performance advantages. In terms of temporal information utilization, the common practice is still 3DConv. Few work has explored a specific temporal modeling mechanism for RSS and even less conducted a discussion of other off-the-shelf methods from the perspective of RSS task.

Considering the above research gaps, this paper conducts a comprehensive discussion from the perspective of RSS for existing time series modeling paradigms, including i) *causal temporal relation modeling*, represented by hidden Markov models (HMM) [25] and recurrent neural network (RNN) family [8, 7, 4]; ii) *parallelized sequence representation modeling*, represented by 3DConv [29] and Transformer [30] family. Based on this, we propose a novel modeling paradigm for radar temporal information utilization. Specifically, this paper makes the following efforts:

i. **A novel temporal modeling paradigm for RSS**. Based on the in-depth analysis and discussion of existing temporal modeling methods, several specific design principles for RSS temporal modeling paradigm are given (§ 2). Following these principles, a novel temporal information learning paradigm for radar spatio-temporal tensors is proposed, *i.e.*, *data-driven temporal information aggregation with learned target-history relations*.

ii. **Temporal relation attentive module (TRAM)**. TRAM is proposed to realize the learning paradigm mentioned above, which is a flexible temporal relation encoding and aggregating module. For encoding, the **t**arget-**h**istory **t**emporal **r**elation **e**ncoder (TH-TRE) is designed, which aims to capture relations between target frame and its historical neighbors. For aggregating, **t**emporal **r**elation **a**ttentive **p**ooling (TRAP) with two forms, *i.e.*, learning in temporal-depth and spatio-temporal, is presented. Both the encoder and aggregator can be independently inserted into arbitrary 2/3DConv-based networks (§ 3).

iii. **Reasonable-scale RSS Network with variable input time lengths**. Processing high-dimensional radar sequence while ensuring the use of temporal information is computationally expensive. To ensure RSS model has the ability of online real-time processing, **t**emporal-**a**ware **RSS** **net**work (TARSS-Net) with moderate parameters is designed based on TRAM, which keeps its parameter scale constant while accepting adjustable time length input. Compared with existing temporal modeling methods, TARSS-Net could effectively avoids the long-term dependence decreasing risk as well as gradient disappearance problem, which shows good segmentation performance and inference efficiency in both multi-view and single-view conditions (§ 3, Appendix C).

iv. **Model performance verification with real-measured data in different detection scenarios**. To verify the scope of application of TARSS-Net, we conduct quantitative experiments on different real-measured large scale radar datasets including CARRADA [22] which is collected from a low cost FMCW ($\approx$ 77GHz) on-board radar in driving scenario and self-collected dataset, KuRALS, recorded from a Kurz-under (Ku) band ($\approx$ 17GHz) radar for UAV surveillance and sea monitoring. Experimental results show TARSS-Net can achieve state-of-the-art (SoTA) performance (§ 4).

## 2   Discussion of Temporal Modeling Paradigms for RSS

In this section, a deep discussion and analysis of current temporal modeling paradigm is conducted. On this basis, the design principles of spatio-temporal encoding suitable for RSS domain are given, and then TRAM paradigm is developed. For better explanation, Fig. 1 intuitively presents the core modeling mechanism of the discussed paradigms.

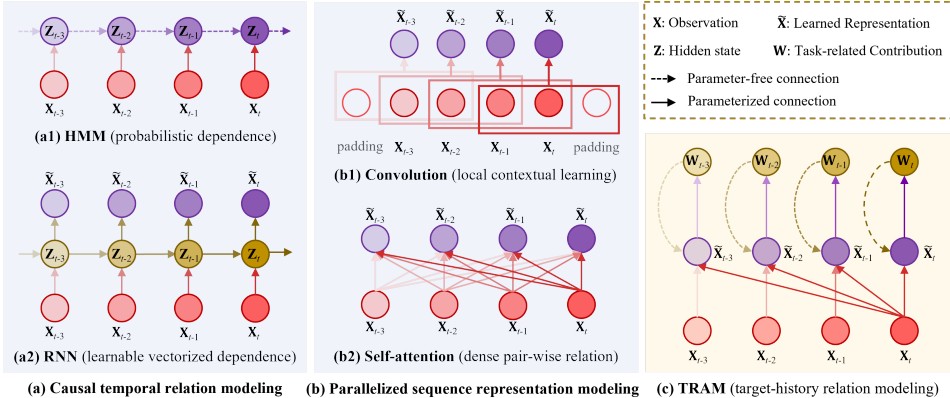

Figure 1: Brief illustration of temporal modeling paradigms.

i. **Causal temporal relation modeling**.

- **HMM** [25] is the classic causal temporal modeling methods, which uses hidden states to model temporal dependencies as shown in Fig. 1-(a1). Based on the assumption of Markov first-order homogeneity and observation independence, it can conveniently use the transition probabilities between hidden states to describe the intrinsic causal relationship of sequential data. However, as a shallow probabilistic model, HMM cannot perform data representation and downstream task prediction end-to-end. Meanwhile, the Markov assumption and discretized encoding of the hidden states also limit its ability to describe long-term dependencies.

- **RNN** [8] also introduces hidden state to describe temporal relation of input sequence similar with HMMs. However, its hidden state is extended to a continuous vector, *i.e.*, real-valued representation projected by a parameterized connection shown in Fig. 1-(a2). In this way, RNN is no longer limited to a probabilistic model, but a learnable one that can be deepened. The gradient descent algorithm can be used to realize end-to-end training of RNN's representation and prediction, which greatly alleviates the problem of HMM's insufficient ability to depict long-term dependencies. With more complex computing units such as LSTM [12, 7] and GRU [4], its representational power can be further enhanced. However, in addition to the resistance caused by gradient dispersion/disappearance under the back-propagation framework, RNNs are sequential causal computing model after all, and their popularity is gradually fading under current main theme of multi-core parallel computing framework.

ii. **Parallelized sequence representation modeling**.

- 3**DConv** [29] uses the concept of local receptive field (LRF) and the design of shared kernel. It can efficiently fulfill encoding and representation of high-dimensional spatio-temporal tensors in a multi-core computing environment as shown in Fig. 1-(b1). Therefore, 3DConv has become the preference for RSS models that require spatio-temporal encoding capabilities [14, 21, 33, 6]. However, due to the limitation of LRF, it cannot achieve temporal-dependence across long-term range while retaining efficient computation and appropriate parameter amount. Meanwhile, an inherent contradiction between 3DConv and RSS is ignored by most of current works, that is, convolution naturally follow the rules of context encoding, which is more suitable for obtaining responses in the center position of LRF. For input sequence, $\{\mathbf{x}_1, \cdots, \mathbf{x}_{1+\tau}, \cdots, \mathbf{x}_{1+2\tau}\} \in \mathbb{R}^{H \times W \times (2\tau+1)}$, performing predictions on $\mathbf{x}_{1+\tau}$ is more of a natural advantage of 3DConv-based models, instead of predicting $\mathbf{x}_{1+2\tau}$ as most RSS required. In terms of pragmatism, performing the prediction at the current instant, *i.e.*, $(1 + 2\tau)$-th time step, is more reasonable, more causal, and more real-time. For this reason, 3DConv might not be optimal for RSS.

- **Transformer** [30] is becoming the backbone of fundamental models in many computing fields [23, 16]. Compared with RNN, it overcomes the problem of parallel computing for handling sequential data; compared with convolution, it breaks the limitation of LRF by preserving the sequence-to-sequence encoding style. The key to Transformer's success lies in self-attention (SA) mechanism. However, its problem is also with SA. By using parameterized connections with different weights, the sequence $\{\mathbf{x}_t\}_{t=1}^{T}$ is abstracted into the forms of $\{\mathbf{q}_t\}_{t=1}^{T}$, $\{\mathbf{k}_t\}_{t=1}^{T}$ and $\{\mathbf{v}_t\}_{t=1}^{T}$, and then the temporal relation between any two primitives is obtained by densely calculating the pairwise

inner product along time dimension, *i.e.*, $\{\mathbf{q}_i \mathbf{k}_j^\top\}_{i=1,j=1}^{T,T}$, resulting the computational complexity of $\mathcal{O}(T^2)$ for temporal relation modeling. This greedy manner is acceptable for data where only time-dimension is a concern, but for spatio-temporal tensors with large spatial ranges, *e.g.*, radar data, the computational cost versus performance gain ratio may not be cost-effective. Moreover, since predictions are required only on current time step, computing resources should not be blindly allocated to each historical time step of input, which would lead to redundancy.

The above analysis motivates this paper to redesign spatio-temporal encoding module for RSS task, and teases out the following 5 design principles:

- Utilizing **parameterized rather than probabilistic connections to characterize the temporal relations** of sequence like Transformers and RNNs do.
- On the premise of making predictions at current time step, the module should **emphasize the use of the current input frame**, *i.e.*, the non-context calculation in time.
- The module should be able to handle **temporal relations in parallel**.
- Considering the high-dimensional spatio-temporal characteristics of radar data, the module should **ensure the efficient learning ability of long-term relationship and keep the parameters appropriately scaled**.
- Due to the coupling of device noise and environmental clutter, radar data would be non-smooth in time dimension. In this way, the module should **consider the contribution of each time step differently** during historical information aggregation.

To take into account the above principles, a novel temporal learning paradigm for RSS is proposed as shown in Fig. 1-(c), *i.e.*, *data-driven temporal information aggregation with learned target-history relations*. The careful designs are as follows:

- The idea of modeling temporal relations with parameterized connections in RNNs and Transformers is preserved, *i.e.*, *temporal relation is explicitly treated as an intermediate embedding obtained by a parameterized layer/block* (relation embeddings mentioned in §§ 3.1).
- The *current/target-history relation encoding mechanism* is introduced (see TH-TRE in §§ 3.1). In this way, the complexity of $\mathcal{O}(T^2)$ for temporal relations in SA is avoided ($\mathcal{O}(T)$ for TRAM), the use of the current frame is emphasized, and the convolution is non-context calculation in time.
- In order to achieve temporal relation encoding in the parallel way, temporal-relation-inception convolution is designed in TH-TRE (TRIC mentioned in §§ 3.1), which is also helpful to *keep model scale constant with variable time length of input sequence*.
- A learnable temporal pooling layer (see TRAP in §§ 3.2) is designed to *measure and reallocate the contribution degree of different target-history relation embeddings*, thus the adverse effects caused by time non-smoothing of radar data quality can be alleviated in a data-driven way.

## 3 Temporal Relation Attentive Model (TRAM)

As illustrated in Fig. 2, the proposed TARSS-Net is based on CAED framework, which consists of basic encoder, TRAM, latent space encoder (LSE) and decoder. For each single-view radar input sequence, the basic encoder is used to generate high-level representations. The LSE is used to align and fuse the high-level semantic features of different views, which is further applied to the single-view decoder to improve its performance. Decoder receives inputs from TRAM and LSE, and finally produces segmentation results on RD and RA perspectives, respectively. As the key of temporal relation learning, TRAM will be detailed in this section, and remaining components can be found in Appendix C. TRAM contains of two components: the **t**arget/current-**h**istorical **t**emporal **r**elation **e**ncoder (TH-TRE) and **t**emporal **r**elation-**a**ware **p**ooling (TRAP).

### 3.1 Target-History Temporal Relation Encoding (TH-TRE)

TH-TRE aims at capturing temporal relations of encoded target frame and its adjacent historical frame features. To achieve this goal, we design the temporal-relation-inception convolution (TRIC) block to handle each target-historical feature pair, as shown in Fig. 3. Given the feature map sequence

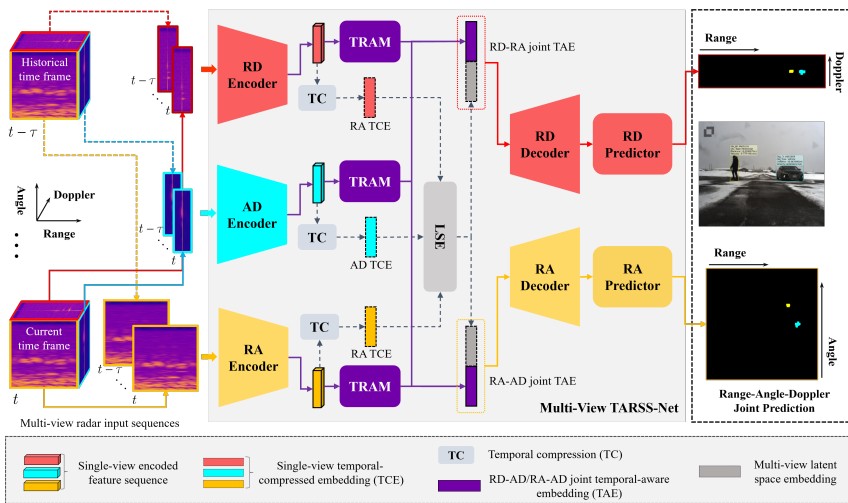

Figure 2: The illustration of multi-view TARSS-Net. The segmentation results for the radar data in RD and RA views, as well as referenced detection results in the camera image are presented for intuitive illustration.

$\{\mathbf{X}_{t-\tau}, \ldots, \mathbf{X}_{t-1}, \mathbf{X}_t\} \in \mathbb{R}^{C \times (\tau+1) \times H \times W}$ obtained from a basic encoder (*e.g.*, RA Encoder), the whole process of TH-TRE can be formalized in Eq. (1). $C$, $(\tau + 1)$, $H$, $W$ are the numbers of channels/depth, time frames, range cells and angle cells, respectively.

$$TH\text{-}TRE\left(\{\mathbf{X}_j\}_{j=t-\tau}^t\right) = \left\{TRIC\left(\mathbf{X}_t, \{\mathbf{X}_i\}_{i=t-\tau}^{t-1}\right) \oplus^{\mathcal{T}} \mathtt{Max}\left(\mathcal{K}_1(\mathbf{X}_t)\right)\right\},$$

$$\text{where } TRIC\left(\mathbf{X}_t, \{\mathbf{X}_i\}_{i=t-\tau}^{t-1}\right) = \left\{\mathcal{K}_2\left(\mathcal{K}_1(\mathbf{X}_t) \oplus^{\mathcal{D}} \mathcal{K}_1(\mathbf{X}_i)\right)\right\}_{i=t-\tau}^{t-1}. \tag{1}$$

Where $\mathcal{K}_1(\cdot) : \mathbb{R}^{C \times H \times W} \to \mathbb{R}^{C \times H_1 \times W_1}$ and $\mathcal{K}_2(\cdot) : \mathbb{R}^{2C \times H_1 \times W_1} \to \mathbb{R}^{C \times H_2 \times W_2}$ are 2D convolution layers, the operator $\oplus^{\mathcal{D}}$ and $\oplus^{\mathcal{T}}$ denotes concatenation on depth and temporal dimension, respectively, $\mathtt{Max}$ is the 2D max-pooling operation with the spatial downsampling rate of 2. In this work, $H_1 = H/2$, $W_1 = W/2$, $H_2 = H/4$, $W_2 = W/4$.

It can be seen that $\mathcal{K}_1(\cdot)$ drags input feature maps of different time frames into a common feature space to ensure representation compatibility [34], and then $\mathcal{K}_2(\cdot)$ takes the enhanced feature maps of each compatible feature pair, *i.e.*, $\mathcal{K}_1(\mathbf{X}_t) \oplus^{\mathcal{D}} \mathcal{K}_1(\mathbf{X}_i)$, as input to learn the relations between target frame and its historical frames. In this way, the associations between target feature maps, $\mathbf{X}_t$, and each of its historical feature maps, $\{\mathbf{X}_i\}_{i=t-\tau}^{t-1}$ is obtained. Finally the outputs from TRIC will be grouped together with $\mathcal{K}_1(\mathbf{X}_t)$ to form the output relation embeddings, *i.e.*, $\{\tilde{\mathbf{X}}_{t-\tau}, \ldots, \tilde{\mathbf{X}}_t\}$ shown in Fig. 3, It is worth noting that convolution kernels in TH-TRE are shared, which helps it accept input with adjustable time length while keeping the number of parameters constant.

## 3.2 Temporal Relation-Aware Pooling (TRAP)

The relation embeddings obtained by TH-TRE block are still in a temporal sequence form. While, for making final predictions on the target frame, the aggregation for generating summarized representation of entire input sequence is required. To this end, a learnable temporal pooling, *i.e.*, TRAP, is proposed. In general, the TRAP block aims at perceiving the contribution degree of each historical frame for prediction task according to the target-history relations, and using these measurements of importance to aggregate the temporal information in each

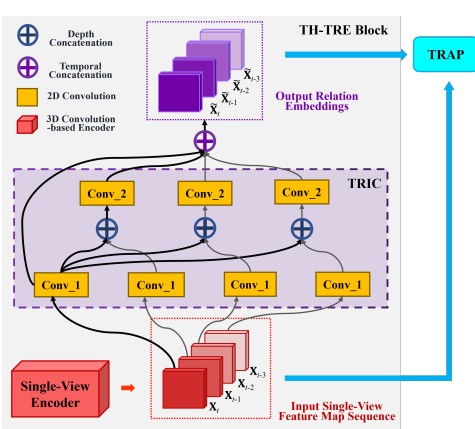

Figure 3: The illustration of TH-TRE.

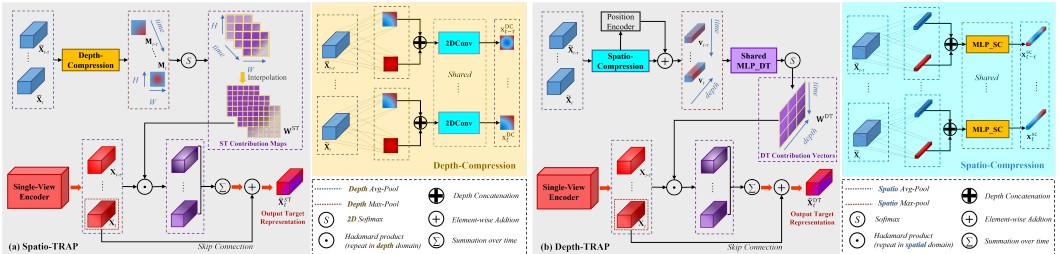

Figure 4: The illustration of two forms of the proposed TRAP block: (a) Spatio-TRAP with Depth-Compression; (b) Depth-TRAP with Spatio-Compression.

single-view radar sequence. However, due to the computational pressure and parameter expansion caused by high-dimensional spatial-time tensors, it is necessary to consider the trade-off of information utilization in each dimension before aggregation. In turn, two forms of TRAP are presented, *i.e.*, Spatio-TRAP and Depth-TRAP. For each form of the TRAP, there are two main steps: compression (discard) and attentive pooling (aggregation).

**Spatio-TRAP.** Spatio-TRAP is performed on the entire spatial domain of input feature maps. Therefore, the importance of temporal relations will be estimated on the spatial space of relation embeddings. As shown in Fig. 4-(a), given the relation embeddings, $\left\{\tilde{\mathbf{X}}_i\right\}_{i=t-\tau}^{t} \in \mathbb{R}^{C\times(\tau+1)\times H_2\times W_2}$, the two main steps of Spatio-TRAP are as follows:

i. *Depth-Compression.* First, all the relation embeddings are compressed by both global depth average-pooling and max-pooling. Then a temporal-shared 2D convolution block is used to generate the contribution map sequence, $\{\mathbf{M}_i\}_{i=t-\tau}^{t} \in \mathbb{R}^{1\times(\tau+1)\times H_2\times W_2}$, for following aggregation step. For each $\mathbf{M}_i$, the process can be formalized as follows,

$$\mathbf{M}_i = \mathcal{K}_2^{\text{ST}}\left(\mathcal{K}_1^{\text{ST}}\left(\text{Avg}^{\text{DC}}(\tilde{\mathbf{X}}_i) \oplus^{\mathcal{D}} \text{Max}^{\text{DC}}(\tilde{\mathbf{X}}_i)\right)\right). \tag{2}$$

Where $\text{Avg}/\text{Max}^{\text{DC}}(\cdot)$ denotes the global depth average/max-pooling operation, $\mathcal{K}_1^{\text{ST}}(\cdot)$ : $\mathbb{R}^{2\times H_2\times W_2} \rightarrow \mathbb{R}^{16\times H_2\times W_2}$ and $\mathcal{K}_2^{\text{ST}}(\cdot)$ : $\mathbb{R}^{16\times H_2\times W_2} \rightarrow \mathbb{R}^{1\times H_2\times W_2}$ are temporal-shared convolution layers with kernel size of $3 \times 3$ and $1 \times 1$, respectively.

ii. *Spatio-Temporal Attentive Pooling*. To modulate the input feature maps in spatio-temporal dimension, a 2D softmax is performed on each map of $\{\mathbf{M}_i\}_{i=t-\tau}^{t}$. Moreover, as deriving from the relation embeddings ($\left\{\tilde{\mathbf{X}}_i\right\}_{i=t-\tau}^{t}$), the contribution maps have a smaller spatial size than $\{\mathbf{X}_i\}_{i=t-\tau}^{t}$, *i.e.*, $[H_2, W_2] = 1/4[H, W]$. Hence a spatial interpolation operation for expanding the size of $\{\mathbf{M}_i\}_{i=t-\tau}^{t}$ is required, so as to ensure the dimension consistency of the following weighted summation and skip connection. Then the whole process of this step can be formalized as

$$\hat{\mathbf{X}}_t^{\text{ST}} = \left\{\sum_{i=t-\tau}^{t} \mathbf{W}_i^{\text{ST}} \odot \mathbf{X}_{i,d}\right\}_{d=1}^{C} + \mathbf{X}_t, \text{where,}$$

$$\mathbf{W}_i^{ST} = \frac{\text{Intp}\left(\{\text{2DSoftmax}(\mathbf{M})\}_i, [H/H_2, W/W_2]\right)}{HW/H_2W_2}. \tag{3}$$

Where, $\hat{\mathbf{X}}_t^{\text{ST}}$ and $\mathbf{W}_i^{\text{ST}}$ are enhanced target representation and spatio-temporal contribution weights of the $i$-th frame. Since the nearest interpolation $\text{Intp}(\cdot, [\cdot, \cdot])$ expands $HW/H_2W_2$ times the spatial size of $\text{2DSoftmax}(\mathbf{M})$, the values of $\mathbf{W}_i^{\text{ST}}$ should be shrank to satisfy $\sum_{i=1,h=1,w=1}^{\tau+1,H,W} \mathbf{w}_{i,h,w}^{\text{ST}} = 1$.

**Depth-TRAP.** Depth-TRAP is performed on the depth of input feature maps. During learning process, features expanded by the depth dimension are of great significance for unit-level classification (semantic segmentation), and they adequately represent semantic information. In this way, Depth-TRAP measures the importance of temporal relations on semantic space of relation embeddings. Similar with the spatio form, Depth-TRAP also requires compression and aggregation steps (see Fig. 4-(b)), which are as follows:

i. *Spatio-Compression*: The input relation embeddings are first compressed in spatial domain by global average-pooling and max-pooling, then these two global pooled sequences are concatenated

and presented to a temporal-shared small MLP network, $\mathcal{G}^{\text{SC}}(\cdot) : \mathbb{R}^{2C} \to \mathbb{R}^{C}$, to obtain the spatio-compressed sequence, $\{\mathbf{x}_i^{\text{SC}}\}_{i=t-\tau}^{t}$. This process can be formalized as follows,

$$\mathbf{x}_i^{\text{SC}} = \mathcal{G}^{\text{SC}}\left(\texttt{Avg}^{\text{SC}}(\tilde{\mathbf{X}}_i) \oplus^{\mathcal{D}} \texttt{Max}^{\text{SC}}(\tilde{\mathbf{X}}_i)\right). \tag{4}$$

Where $\texttt{Avg/Max}^{\text{SC}}(\cdot)$ denotes global average/max-pooling operation on spatial domain, with a kernel size of $H_2 \times W_2$.

ii. *Depth-Temporal Attentive Pooling*: This step calculates the contribution vector for each spatial-compressed temporal relation embedding, $\mathbf{x}_i^{\text{SC}}$, through depth dimension, which achieves the importance measurement of temporal relations in semantic space. Then, with these contribution vectors, a weighted summation along with a skip connection is conducted to aggregate each single-view encoding feature sequence, $\{\mathbf{X}_i\}_{i=t-\tau}^{t}$, and therefore the enhanced target representation, $\hat{\mathbf{X}}_t^{\text{DT}} \in \mathbb{R}^{C \times H \times W}$ is obtained. The contribution vectors can be obtained as follows,

$$\mathbf{W}^{\text{DT}} = \texttt{Softmax}\left(\{\mathcal{G}^{\text{DT}}(\mathbf{v}_i)\}_{i=t-\tau}^{t}\right) \text{ where,}$$

$$\mathbf{v}_i = \{x_{i,d}^{\text{SC}} + p_{i,d}\}_{d=1}^{C}, \ p_{i,d} = \begin{cases} p_{i,d} = 0.1\sin\left(i/10^{8(d/2)/C}\right), \text{ if } d \bmod 2 = 0; \\ p_{i,d} = 0.1\cos\left(i/10^{8((d-1)/2)/C}\right), \text{ if } d \bmod 2 = 1. \end{cases} \tag{5}$$

Where $\mathcal{G}^{\text{DT}}(\cdot) : \mathbb{R}^{C} \to \mathbb{R}^{C}$ denotes the temporal-shared MLP, and $p_{i,d}$ is the depth-temporal position encoding result following self-attention [30] for each element of $\{\mathbf{x}_i^{\text{SC}}\}_{i=t-\tau}^{t}$. Similar to Spatio-TRAP, the contribution vectors are scaled by softmax to jointly modulate the input feature maps in both depth and time dimensions. Then the scaled vectors, $\mathbf{W}^{\text{DT}} \in \mathbb{R}^{C \times (\tau+1)}$, are used to aggregate information from input feature maps as follows,

$$\hat{\mathbf{X}}_t^{\text{DT}} = \left\{\sum_{i=t-\tau}^{t} \mathbf{W}_i^{\text{DT}} \odot \mathbf{x}_{i,h,w}\right\}_{h=1,w=1}^{H,W} + \mathbf{X}_t. \tag{6}$$

Where the operator $\odot$ denotes Hadamard product, and $\mathbf{x}_{i,h,w} \in \mathbb{R}^{C}$ is the feature in each spatio-temporal position of input feature maps $\{\mathbf{X}_i\}_{i=t-\tau}^{t}$ obtained from a basic encoder. Finally, the output target representation, $\hat{\mathbf{X}}_t$, enhanced by the TRAM will be presented to the corresponding decoder.

## 4  Experiments

### 4.1  Datasets and training setup

Three datasets have been used to valid the performance of TARSS-Net including: **CARRADA**, which contains multi-view annotated radar recordings (RAD tensors) for 4 categories of objects in driving scenarios under different weathers conditions; **CARRADA-RAC** [35], which is an improved version of CARRADA with calibrations on RA view; **KuRALs**, which is a real-measured dataset with refined annotation collected by a Ku-band surveillance radar and includes various typical targets. The training setup for TARSS-Nets and other compared SoTA networks strictly follows the consistent configuration, including hardware platform, hyper-parameter settings and evaluation metrics, and the number of input frames for TARSS-Nets is 5 by default ($\tau = 4$). See Appendix D for more details.

### 4.2  Comparisons with State-of-The-Art Methods

The most widely used RSS dataset, CARRADA, is used to comprehensively compare the performance of TARSS-Net with existing RSS methods. CARRADA-RAC and KuRALS dataset are utilized to explore the generalization and stability of TARSS-Net, on which only representative methods are compared, *i.e.*, TMVA-Net (temporal-based model) and PKCIn-Net (current SoTA). TARSS-Net_S and TARSS-Net_D denote the TARSS-Net with Spatio-TRAP and Depth-TRAP, respectively.

i. **CARRADA**. The overall results on the test subset are listed in Table 1. Generally, the models that explicitly consider the temporal dynamics in radar signals, *e.g.*, TMVA-Net [21], can obviously achieve better performance than other convolution-based models [20, 28, 2, 14, 6]. By introducing SA

mechanism in Transformers [30, 18] into convolution frameworks, TransRadar [5], T-RODNet [13] (the RD scores are not provided in the literature) and TransRSS [36] (the code is not public in the literature) further enhance the representation ability in spatial dimension, thus pushing the performance of RSS to a new height. However, due to the explosion of parameter scale caused by the high-dimensional radar tensors, none of them use SA in temporal domain. Compared with the existing RSS methods using temporal information, TARSS-Nets with the well-designed temporal learning paradigm exactly follows the 5 design principles mentioned in § 2 and achieves the best overall performance (average metrics for both views) with reasonable parameter scales. Unfortunately, its RA performance is not so prominent, as the RA data does not change much over time or poor angular resolution, resulting in less obvious temporal information that can be exploited. Specially, PKCIn-Net [35] does not belong to the category of temporal information modeling, but its better performance in the RA view shows that it is helpful to introduce the classical radar signal processing theory into RSS models, which provides a direction for the future improvement of TARSS-Nets.

ii. **CARRADA-RAC**. Models listed in Table 1 fail to aquire the ideal performance on the RA view. One main reason is the poor angular resolution of common low-cost FMCW radars. CARRADA-RAC [35] has calibrated the RA annotations, which could give a more realistic performance comparison for different models. As shown in Table 2, the global performance are improved after annotation refinement, especially in RA view and TARSS-net still achieves the performance of SoTA.

iii. **KuRALS**. This dataset is different from CARRADA in terms of radar systems and detection scenarios, which could complement testing the generalization and stability of models. Experiments in Table 3 show that TARSS-Net cloud maintain its superior performance in different application scenarios. The model marked with $^{sv}$ has been modified to a single-view version as needed.

Table 1: Comparisons with SoTA RSS networks.

| Method | #Param. | RD-View (%) | | RA-View (%) | |
|---|---|---|---|---|---|
| | | mIoU | mDice | mIoU | mDice |
| FCN[20] | 134.3M | 54.7 | 66.3 | 34.5 | 40.9 |
| U-Net[28] | 17.3M | 55.4 | 68.0 | 32.8 | 38.2 |
| DeepLabv3+[2] | 59.3M | 50.8 | 61.6 | 32.7 | 38.3 |
| RSS-Net[14] | 10.1 M | 32.1 | 36.9 | 32.1 | 37.8 |
| RAMP-CNN[6] | 106.4 M | 56.6 | 68.5 | 27.9 | 30.5 |
| MV-Net[21] | **2.4 M** | 29.0 | 32.8 | 26.8 | 28.5 |
| MVA-Net[21] | 4.8 M | 53.5 | 65.3 | 37.1 | 44.8 |
| TMVA-Net[21] | 5.6 M | 56.1 | 68.0 | 37.7 | 46.2 |
| TransRadar[5] | 4.9M | 57.2 | 69.1 | 39.9 | 49.5 |
| T-RODNet[13] | 162.0M | - | - | **43.5** | 53.6 |
| TransRSS[36] | - | 60.4 | 73.0 | 43.0 | **53.8** |
| PKCIn-Net[35] | 6.3M | 60.7 | 72.6 | 43.1 | 53.7 |
| TARSS-Net_S | 6.2 M | 62.1 | 73.8 | 41.6 | 51.2 |
| TARSS-Net_D | 6.3 M | **63.4** | **75.2** | 41.4 | 51.3 |

Table 2: Performance on CARRADA-RAC.

| View | Method | #Param. | mIoU | mDice |
|---|---|---|---|---|
| RD | **TMVA-Net** | **5.6M** | 59.7% | 69.9% |
| | **PKCIn-Net** | 6.3M | 60.6% | 72.4% |
| | **TARSS-Net_S** | 6.2M | 62.5% | 74.3% |
| | **TARSS-Net_D** | 6.3M | **62.8%** | **74.6%** |
| RA | **TMVA-Net** | **5.6M** | 46.6% | 57.9% |
| | **PKCIn-Net** | 6.3M | 47.3% | **58.7%** |
| | **TARSS-Net_S** | 6.2M | 45.8% | 56.1% |
| | **TARSS-Net_D** | 6.3M | **47.4%** | **58.7%** |
| Global | **TMVA-Net** | **5.6M** | 53.2% | 63.9% |
| | **PKCIn-Net** | 6.3M | 54.0% | 65.6% |
| | **TARSS-Net_S** | 6.2M | 54.1% | 65.2% |
| | **TARSS-Net_D** | 6.3M | **55.1%** | **66.7%** |

## 4.3 Ablation Study

Three sets of ablation experiments are conducted to verify the effectiveness of proposed TRAM module and its two independent blocks, TH-TRE and TRAP. Depth-TRAP is used as default.

**Ablation Experiments on TRAM**. As control experiments, two baseline models are designed by incorporation different mechanisms of temporal information utilization to TARSS-Net: i) *Baseline-A*: replacing TRAM from TARSS-Net with a single temporal global average/max-pooling (GAP/GMP) layer; ii) *Baseline-B*: replacing TRAM from TARSS-Net with temporal aggregation layers designed in TMVANet [21], which consists of stacked 3DConvs. Results show in Table 4. Baseline-B achieves better performance on both

Table 3: Performance comparison on KuRALS.

| Method | #Param. | RD View | |
|---|---|---|---|
| | | mIoU | mDice |
| FCN | 134.3M | 50.4% | 59.4% |
| U-Net | 17.3M | 52.4% | 60.1% |
| DeepLabv3+ | 59.3M | 52.6% | 61.8% |
| TMVA-Net$^{sv}$ | **1.2M** | 52.9% | 63.1% |
| PKCIn-Net$^{sv}$ | **1.2M** | 56.7% | 65.9% |
| TARSS-Net_S$^{sv}$ | **1.2M** | 53.2% | 63.8% |
| TARSS-Net_D$^{sv}$ | 1.3M | **58.4%** | **67.1%** |

views than Baseline-A, which indicates that parameterized learnable aggregation has obvious advantages over parameter-free (max or avg) aggregation. While by using the proposed TRAM, TARSS-Net can obviously exceed 3DConv-based Baseline-B on almost every evaluation metric, which fully demonstrates the effectiveness of the proposed temporal-aware learning paradigm for RSS networks.

**The Effectiveness of TH-TRE**. TH-TRE aims to capture temporal relations of target frame and its historical frames. To verify its effectiveness, two more contrast models are presented: i) *Baseline-A w/ GAP & TH-TRE*: adding TH-TRE block to Baseline-A w/ GAP, in which a single temporal GAP layer is used to aggregate the temporal relation embeddings obtained from TH-TRE; ii) *TARSS-Net w/o TH-TRE*: TARSS-Net without explicitly learning temporal relations of the target frame and its historical frames. Here, the TRIC block is replaced with a regular 3DConv block, which contains two convolution layers with the same kernel size of $1 \times 3 \times 3$ and stride of $(1, 2, 2)$. As shown in Table 5, the addition of TH-TRE block is beneficial to improve RSS performance of both Baseline-A w/ GAP and TARSS-Net w/o TH-TRE. This block realizes current/target-history relation encoding under the premise of maintaining temporal order. In this way, TH-TRE emphasizes the use of current frame and achieve more computationally efficient and powerful temporal encoding capacity than simple temporal compression mechanism such as GAP and 3D convolution.

Table 4: Ablation experimental results on TRAM.

| Method | RD-View (%) | | | | RA-View (%) | | | | Global (%) | | | |
|---|---|---|---|---|---|---|---|---|---|---|---|---|
| | Prec. | Recall | mIoU | mDice | Prec. | Recall | mIoU | mDice | Prec. | Recall | mIoU | mDice |
| Baseline-A w/ GAP | 61.8 | 76.3 | 55.4 | 67.8 | 43.5 | 47.1 | 36.4 | 44.7 | 52.7 | 61.7 | 45.9 | 56.3 |
| Baseline-A w/ GMP | 68.6 | 68.2 | 50.8 | 62.1 | 49.3 | 42.9 | 36.1 | 42.7 | 59.0 | 55.6 | 43.5 | 52.4 |
| Baseline-B | 63.6 | 74.8 | 56.1 | 68.0 | 44.2 | 51.6 | 37.7 | 46.2 | 53.9 | 63.2 | 46.9 | 57.1 |
| **TARSS-Net w/ TRAM** | **70.9** | **80.6** | **63.4** | **75.2** | **56.1** | 50.0 | 41.4 | **51.3** | **63.5** | **65.3** | **52.4** | **63.3** |

Table 5: Ablation experimental results on TH-TRE.

| Method | RD-View (%) | | | | RA-View (%) | | | | Global (%) | | | |
|---|---|---|---|---|---|---|---|---|---|---|---|---|
| | Prec. | Recall | mIoU | mDice | Prec. | Recall | mIoU | mDice | Prec. | Recall | mIoU | mDice |
| Baseline-A w/ GAP | 61.8 | 76.3 | 55.4 | 67.8 | 43.5 | 47.1 | 36.4 | 44.7 | 52.7 | 61.7 | 45.9 | 56.3 |
| Baseline-A w/ GAP & TH-TRE | 68.9 | 79.3 | 60.6 | 72.3 | 50.9 | 50.2 | 40.4 | 49.7 | 59.9 | 64.8 | 50.5 | 61.0 |
| **TARSS-Net w/o TH-TRE** | 69.9 | 79.2 | 61.6 | 73.7 | 53.2 | 50.1 | 40.4 | 49.8 | 61.6 | 64.7 | 51.0 | 61.8 |
| **TARSS-Net w/ TH-TRE** | **70.9** | **80.6** | **63.4** | **75.2** | **56.1** | 50.0 | **41.4** | **51.3** | **63.5** | **65.3** | **52.4** | **63.3** |

Table 6: The effects of Depth/Spatio-TRAP block.

| Method | RD-View (%) | | | | RA-View (%) | | | | Global-View (%) | | | |
|---|---|---|---|---|---|---|---|---|---|---|---|---|
| | Prec. | Recall | mIoU | mDice | Prec. | Recall | mIoU | mDice | Prec. | Recall | mIoU | mDice |
| Baseline-A_G | 61.8 | 76.3 | 55.4 | 67.8 | 43.5 | 47.1 | 36.4 | 44.7 | 52.7 | 61.7 | 45.9 | 56.3 |
| Baseline-A_S | 69.5 | 78.5 | 61.0 | 73.3 | 50.8 | 49.4 | 38.8 | 47.4 | 60.2 | 64.0 | 49.9 | 60.4 |
| Baseline-A_D | 69.9 | 79.2 | 61.6 | 73.7 | 53.2 | 50.1 | 40.4 | 49.8 | 61.6 | 64.7 | 51.0 | 61.8 |
| **TARSS-Net_G** | 68.9 | 79.3 | 60.6 | 72.3 | 50.9 | 50.2 | 40.4 | 49.7 | 59.9 | 64.8 | 50.5 | 61.0 |
| **TARSS-Net_S** | 70.4 | 79.1 | 62.1 | 73.8 | 53.6 | 50.6 | 41.6 | 51.2 | 62.0 | 64.9 | 51.9 | 62.5 |
| **TARSS-Net_D** | **70.9** | **80.6** | **63.4** | **75.2** | **56.1** | 50.0 | 41.4 | **51.3** | **63.5** | **65.3** | **52.4** | **63.3** |

**The Effectiveness of TRAP**. TRAP aims to measure and allocate the contribution degree of different target-history relation embeddings before aggregation. Several models are evaluated to valid its effectiveness: (i) *Baseline-A_G*: Baseline-A with GAP; (ii) *Baseline-A_S/D*: Baseline-A with Spatio/Depth-TRAP block; (iii) *TARSS-Net_G*: replacing TRAP block from TARSS-Net with GAP; (iv) *TARSS-Net_S/D*: TARSS-Net with Spatio/Depth-TRAP block. The results listed in Table 6 show that either form of TRAP block can aggressively improve the performance of both Baseline-A model and TARSS-Net.

This illustrates that different history frame has unequal closeness with target frame, and TRAP would be an efficiently way for reassigning weights to temporal relations aggregation, which effectively alleviates the negative influence on RSS caused by temporal unsmoothness of low-quality radar data. In addition, Depth-TRAP-based models always obtain better results than Spatio-TRAP-based models. Such performance gap may caused by the aggressive compression of the semantic information (reflected in the channel dimension of features) in Spatio-TRAP block. As mentioned in §§ 3.2, Depth-TRAP pays more attention to the semantic features obtained from the TH-TRE block, and such consideration makes the network generate more informative representations than Spatio-

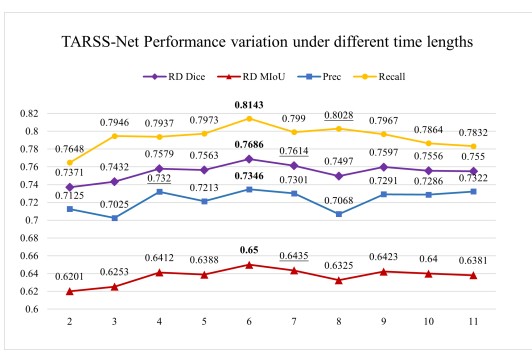

Figure 5: TARSS-Net performance (RD view) variation on CARRADA using input sequence with different numbers of frames.

TRAP. Therefore, we argue that maintaining more semantic information might be an effective way to further improve Spatio-TRAP, which is worth of investigating in the future work.

### 4.4 Effect of input time length on TARSS-Net performance

The RD performance of TARSS-Net_D increases with time length of inputs grows and achieves the optimum at 6 frames, *i.e.*, $\tau = 5$, as presented in Fig. 5. With the careful design of temporal modeling paradigm, TARSS-Net can handle arbitrarily long sequence relations without increasing parameter scale, which is significant to find the optimal duration of temporal dependencies not only for RSS tasks but also for all temporal relation modeling methods.

### 4.5 Real-time performance comparison

In addition to model scale, the computational complexity and inference speed of the RSS methods are also important for deploying applications, which is measured by multiply–accumulate operations (MACs) and frames per second (FPS), respectively. All the real-time performance shown in this section are obtained on a single RTX 3090 GPU.

Among existing RSS models, TMVA-Net is the SoTA one for multi-view temporal relation learning. As far as our knowledge, none of existing methods applied Transformer or SA on the time dimension in RSS task because of the exploding number of parameters. In order to deeply explore the real-time performance of different temporal relationship modeling methods, we introduce vision Transformer (ViT) into the basic architecture of TARSS-Net, which forms a ViT-based baseline method, called Vit-based-Net. The left part of Table 7 compares the real-time performance of TARSS-Net with TMVA-Net and Vit-based-Net with multi-view inputs. All methods take 5 consecutive time frames as input. It can be seen that directly introducing Transformer in temporal relationship modeling of radar data is inefficient, which would consume more computation source but obtaining lower RSS performance. TARSS-Net consumes more computation and inference time than TMVA-Net in temporal relation learning, but achieves better RSS accuracy. In addition, the inference speed of 23 FPS in TARSS-Net_D is also sufficient for real-time applications.

Table 7: Real-time performance (MV: multi-view; SV: single-view).

| Method | Inputs | Params (M) | MACs (G) | FPS | mIoU (%) | mDice (%) | Inputs | Params (M) | RD-View MACs(G) | RD-View FPS | RA-View MACs(G) | RA-View FPS |
|---|---|---|---|---|---|---|---|---|---|---|---|---|
| TMVA-Net | MV | 7.2 | **119.5** | **66** | 46.9 | 57.1 | SV | **1.2** | 11.3 | **250** | 36.6 | **250** |
| Vit-based-Net | MV | 27 | 449 | 12 | 38.1 | 44.5 | SV | 3.6 | **1.8** | 59 | **7.0** | 55 |
| **TARSS-Net_S** | MV | **6.2** | 197.6 | 35 | 51.9 | 62.5 | SV | **1.2** | 13.3 | 181 | 40.4 | 143 |
| **TARSS-Net_D** | MV | 6.3 | 175.4 | 23 | **52.4** | **63.3** | SV | **1.2** | 13.3 | 111 | 40.4 | 112 |

The right part of Table 7 compares real-time performance of several temporal relationship learning models in single view. That is, the results for RD and RA views are measured separately by the corresponding single-view network versions. In terms of real-time performance, TMVA-Net still has the fastest inference speed. But Vit-based-Net achieves the worst inference speed with the most number of parameters. However, it is not to be overlooked that TRASS-Net still leads the pack in terms of RSS performance with competitive real-time performance.

More supplementary experiments are presented in Appendix E, including ablatioin experiment of AD encoding branch, class-wise performance comparisons, features visualization of core process in TARSS-Net and some example visualization of segmentation results.

## 5 Conclusions

This work focuses on exploiting temporal information in radar signals to enhance the representation capacity of multi-view RSS network. Firstly, the advantages and disadvantages of existing temporal modeling methods in RSS domain were deeply discussed, and on this basis, the design principles of RSS spatio-temporal encoding methods were introduced. Based on the principles, a flexible temporal-aware learning module, TRAM, and TARSS-Net based on TRAM is proposed, which follows the novle temporal learning paradigm, *i.e.* data-driven temporal information aggregation with learned target-history relations. Experiments fully verifies the superiority of TARSS-Net through SoTA methods comparison on three datasets, ablation experiments, performance under variation input time length, as well as its real-time performance.

## Acknowledgements

This work is supported by National Natural Science Foundation of China (No. 62206258).

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

# Appendix

Following parts are introduced in this appendix: (i) the input streams for our network, *i.e.*, radar signal processing for multi-view range-angle-Doppler (RAD) tensors; (ii) more detailed contrastive analysis of present deep learning-based sequential models, *e.g.*, Self-Attention [30], RNN family [8, 7, 4], and our temporal-aware learning mechanism; (iii) additional descriptions of our TARSS-Net and TRAM module; (iv) more detailed description of datasets and training setup; (v) additional experiments including ablation of AD Encoding branch, feature visualization and some example visualization of segmentation results; (vi) limitations and social impacts.

## A  From Radar Signals to Multi-View Representations

As mentioned in §. 1, most CAED-based radar detection/segmentation methods use FFTs as signal processing front-end. Despite the loss of fine-grained temporal information, FFT enables pure time domain radar echoes to be expressed in spatial (range and angle) and Doppler. At the same time, it can provide more intuitive and structured input to the model. Therefore, we also use such classic signal processing to aquire the multi-view RAD tensors.

Considering one-frame radar signals received from an FMCW radar, it is composed of multiple chirps from multiple antennas and can be denoted as $\{\mathbf{Chirp}_i^{(j)}\}_{i=1,j=1}^{N_D,N_A}$, where $N_D$ and $N_A$ denote the numbers of chirps and antennas, respectively. Then Range-FFT, $\mathcal{F}_R(\cdot)$, is conducted on each chirp, $\mathbf{Chirp}_i^{(j)}$, to obtain the DFT (Discrete FT) results, then we can get DFT tensor as follows:

$$\{\mathcal{F}_R(\mathbf{Chirp}_i^{(j)})\}_{i=1,j=1}^{N_D,N_A} = \{\mathbf{M}_R^{(j)}\}_{j=1}^{N_A} \in \mathbb{R}^{N_R \times N_D \times N_A}. \tag{S1}$$

Where $\mathbf{M}_R^{(j)}$ denotes the Range-DFT matrix for $j$-th antenna chirps, $N_R$ is the sampling number of each chirp. Please note that we only consider the real part of FFT results for simplification. Then for each row of $\mathbf{M}_R^{(j)}$, the Doppler-FFT, $\mathcal{F}_D(\cdot)$, is conducted to get the Doppler-DFT matrix as follows:

$$\{\mathcal{F}_D(\mathbf{M}_R^{(j)}[k,:])\}_{k=1}^{N_R} = \mathbf{M}_{RD}^{(j)} \in \mathbb{R}^{N_R \times N_D}. \tag{S2}$$

Then we can group these DFT matrixes of all antennas to form the second DFT tensor, $\{\mathbf{M}_{RD}^{(j)}\}_{j=1}^{N_A} = \mathbf{M}_{RD}$. Finally, the Angle-FFT, $\mathcal{F}_A(\cdot)$, is performed on $\mathbf{M}_{RD}$ along the antenna dimension to get final RAD tensor:

$$\{\mathcal{F}_A(\mathbf{M}_{RD}[i,k,:])\}_{i=1,k=1}^{N_R,N_D} = \mathbf{M}_{RAD} \in \mathbb{R}^{N_R \times N_D \times N_A}. \tag{S3}$$

In this work, $\{N_R,\ N_A,\ N_D\} = \{256,\ 256,\ 64\}$. Obviously, even a single frame of $\mathbf{M}_{RAD}$ is also quite dense and bulky for deep models. Further compression is needed to obtain affordable input streams in different frequency domains. Therefore a 2D-based multi-view compressing method is adopted in [21], *i.e.*, averaging over different frequency domains. Taking angle frequency domain as an example, 3D RAD tensor would be compressed as 2D RD view representation as follows:

$$\mathbf{X}_{RD}[i,k] = 10 * \log\left(\frac{1}{N_A}\sum_{j=1}^{N_A}|\mathbf{M}_{[i,j,k]}|^2\right). \tag{S4}$$

Using such processing, the multi-view representations of our models, $\{\mathbf{X}_{RD}, \mathbf{X}_{AD}, \mathbf{X}_{RA}\}$ of one-frame radar tensor can be obtained. And the input scale is agressively reduced from $256 \times 256 \times 64$ to $256 \times 64 + 256 \times 64 + 256 \times 256$.

## B  Contrastive Analysis of TRAM and Other Sequential Models

This section complements the discussion about Temporal Information Utilization in § 2.

For present, one of the important problems is insufficient use of temporal information or clues in radar data. Present works [1, 14, 6, 21, 33] mainly focused on providing a complete solution to radar scene understanding tasks, while spatial-temporal information is relatively simply introduced by some proven techniques in other fields, *e.g.*, 3D convolution or its variant deformable convolution. That is

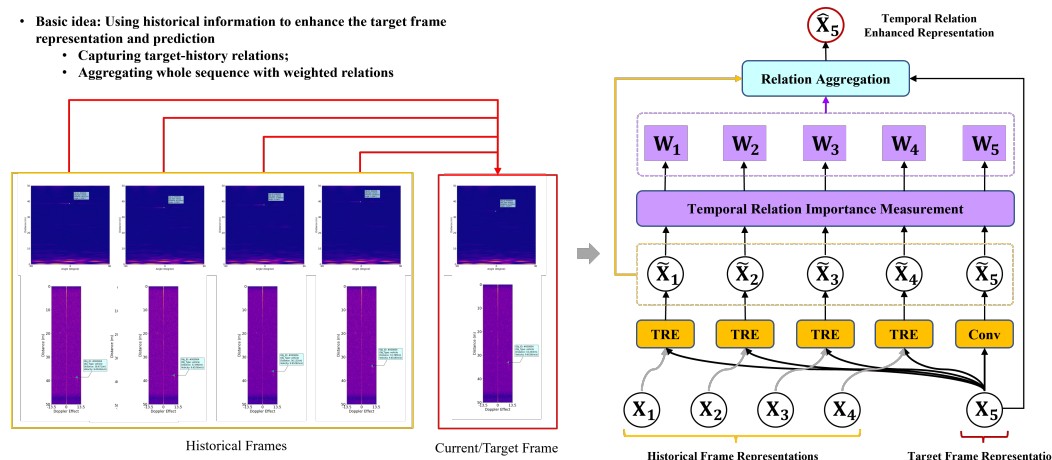

Figure S1: The basic learning paradigm (left) and implementation of proposed TRAM (right). **TRE** in the yellow boxes denote TH-TRE; **Temporal Relation Importance Measurement** relates to the attention weights calculation in TRAP block, and **Relation Aggregation** relates to weighted summation and skip connection in TRAP block.

to say, temporal modeling only plays a limited role in these works. However, in this work, we dedicate to this temporal information utilizing problem, and attempt to define a general learning paradigm: **capturing temporal relations first and then aggregating them by a task-driven manner**. These relations are extracted from target frame and its historical neighbors in an explicit way, then a learnable relation aggregator would estimate the importance of those relations and group them adaptively into a single representation according to the task. It is different from previous works which merely rely on convolutions and learning-free aggregating operators, *e.g.*, max/avg-pooling.

As mentioned in § 1, the existing temporal modeling methods can be summarized into two categories: i) Causal temporal relation modeling, represented by HMM and RNN family; ii) Parallelized sequence representation modeling, represented by 3DConv and Transformer (SA) family. Despite the advantages of these methods, directly using RNN and SA would introduce other problems. For RNNs, the dependencies between elements are captured by a recursive dependency calculation manner. That is to say, to produce an output, $\mathbf{O}_t$ at $t$-th frame, the former one $\mathbf{O}_{t-1}$ should be generated first. Such recursive dependency encoding mechanism could not fully enjoy the convenience of parallel computing. For SA, the dependencies between elements are captured across each pair of elements in the input sequence. However, extracting all context relations could cause redundancy. Since our task is to make prediction on target frame, it is reasonable to pay more attention to target-history pairs instead of each pair. Moreover, SA is more friendly to MLP-based frameworks, but for convolutional networks, its multi-scale spatial encoding capacity would be restricted to some extent since there is a trade-off between information compression and parameter scale when flattening the spatio-structured feature maps into a vector. Comparatively speaking, the proposed TRAM module well overcomes the above shortcomings. It can capture the temporal relationship of variant input time length without increasing scale of network, and at the same time retain the advantage of parallel computing.

To intuitively show our basic idea and parallel computation of proposed TRAM, we drew Fig. S1. One implication can be seen in the figure, that the importance measurement can be regarded as a soft-selection for target-history relations. It actually drives the model to focus on the useful part of history information. That is to say, if one can decide which historical frames contribute most to target frame representation and prediction, it would be easier to design a model with a longer temporal dependence and less learnable parameters, since history information with less importance could be ignored directly. And we argue that *filtering useful historical information in advance* is also worth to investigate in the future, so that the model can "see more history", while keeping a reasonable parameter scale.

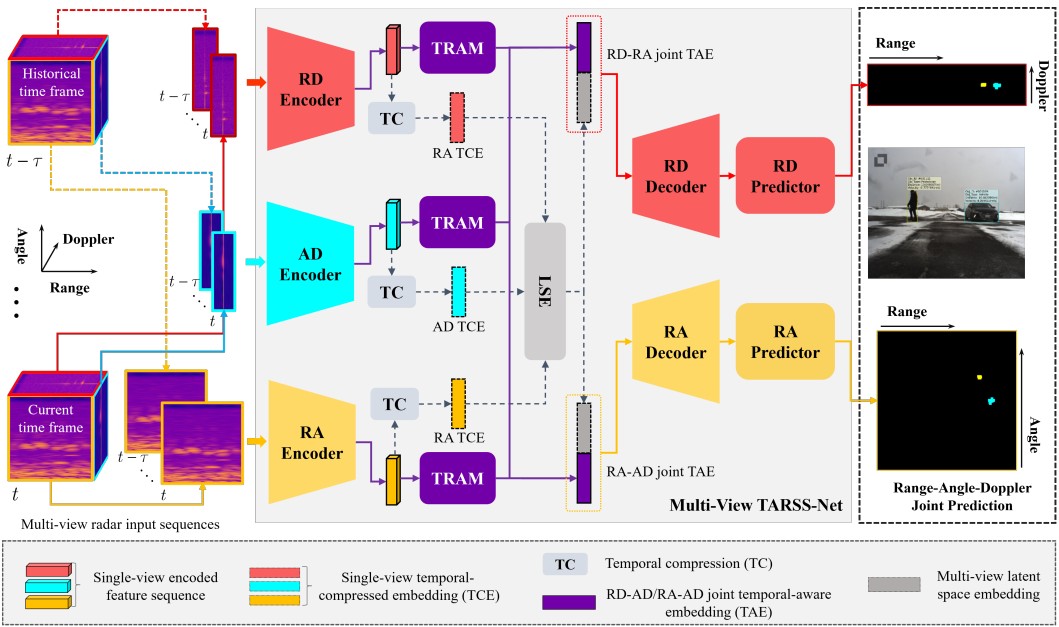

Figure S2: The simplified workflow of TARSS-Net. The segmentation results for the radar data in RD and RA views, as well as referenced detection results in the camera image are presented for intuitive illustration. ASPP module is omitted for simplicity.

## C    Additional Descriptions of TARSS-Net

### C.1    Detailed description of TARSS-Net framework

As illustrated in Fig. S2, the proposed TARSS-Net is based on CAED framework. In addition to TRAM, some core components are introduced as follows.

**Basic encoder**. For each single-view radar input sequence, the basic encoder is used to generate high-level representations, and multi-view radar sequence can be handled by performing these encoders in parallel. All three single-view encoders are designed with the same structure for simplicity. Each encoder is mainly composed of two 3D convolution blocks with kernel size of $1 \times 3 \times 3$, which shares convolution kernel learning of each frame on the time dimension, so as to realize feature interaction between all historical and target frames. **It is worth noting that this is different from using 2D convolutions, which do not enable parameter sharing and feature interaction in dimensions outside the 2D plane.** To further compress the spatial size of feature maps, each convolution block follows a spatial max-pooling layer.

**Latent Space Encoder (LSE)**. The basic encoder and TRAM are performed on each single-view of the radar sequence. However, one problem in multi-perspective learning structure is how to extract the rich knowledge from different information flows, and use this knowledge to improve the performance of the model on each local sub-task or global task. For this purpose, a multi-view shared learner, LSE is added to our TARSS-Net. The inputs of LSE are temporal-compressed embeddings (TCEs) obtained by temporal compression (TC) module, which consisted with a global temporal average-pooling layer and a 2D convolution layer with $1 \times 1$ kernels. Two shared $1 \times 1$ 2D convolution layers are stacked to form the LSE, which will project the TCEs into a common latent space to get a uniform embedding. Since multi-view inputs are used, the $1 \times 1$ 2D convolution layers are chosen instead of the fully connected layers for reducing the model parameters.

**Decoder**. After encoder, the features obtained from TRAM and LSE could comprehensively consider single-view temporal relationship and multi-view relationship alignment, which will be fed into decoder. Only two branches for RD and RA views are designed in decoder, which is enough to complete target location. By the way, this is also the reason why existing RSS datasets is not labeled on AD view with groundtruth. Each of decoders consists of the combination of 2D deconvolutions and convolutions as in [21].

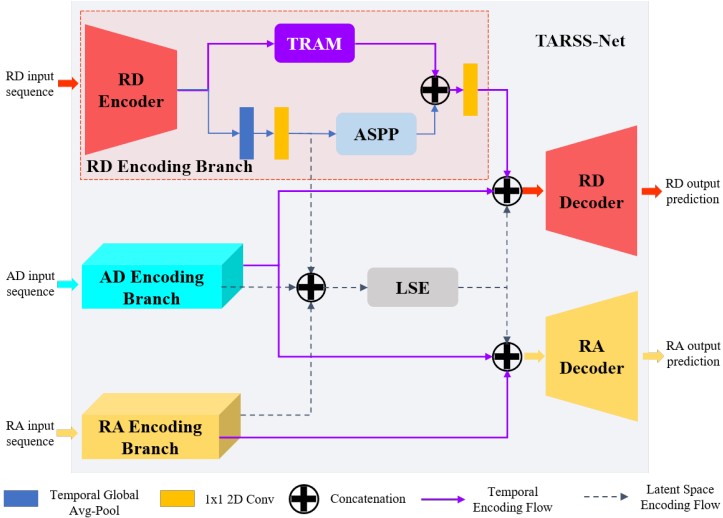

Figure S3: The overall framework of TARSS-Net: details of the encoding branch for RD-view is given as an example.

In general, Fig. S3 gives an intuitive illustration of the overall structure of our TARSS-Net. The encoding branch would take multi-view radar tensor (*i.e.*, RD, AD, and RA) streams within several time frames as input, and generate the unified representation of these streams in a common latent space. Thus the encoding branch is used to learn representations and the decoding branch is used to interpret those representations. Besides each basic encoder, the proposed TRAM and ASPP modules aim to enhance the representation by capturing and measuring target-history relations and injecting multi-scale spatial information from each single view. Using these enhanced representations as input, the decoder would make cell-wise predictions on RD and RA views, respectively.

## C.2    More in-depth discussion of TH-TRE block

The TRIC (temporal-relation-inception convolution) block proposed in TH-TRE (§ 3.1) can also be implemented with the 3D convolution. Firstly, let's recall the introduced implementation of TH-TRE:

$$
\begin{aligned}
TH\text{-}TRE\left(\{\mathbf{X}_j\}_{j=t-\tau}^{t}\right) &= \left\{TRIC\left(\mathbf{X}_t,\ \{\mathbf{X}_i\}_{i=t-\tau}^{t-1}\right)\oplus^{\mathcal{T}}\texttt{Max}\left(\mathcal{K}_1(\mathbf{X}_t)\right)\right\}, \\
\text{where } TRIC\left(\mathbf{X}_t,\ \{\mathbf{X}_i\}_{i=t-\tau}^{t-1}\right) &= \left\{\mathcal{K}_2\left(\mathcal{K}_1(\mathbf{X}_t)\oplus^{\mathcal{D}}\mathcal{K}_1(\mathbf{X}_i)\right)\right\}_{i=t-\tau}^{t-1}.
\end{aligned}
\tag{S5}
$$

Where $\mathcal{K}_1(\cdot)$ and $\mathcal{K}_2(\cdot)$ are 2D convolution layers. The 3D-based implementation of TRIC can be conducted as follow:

$$
\begin{aligned}
TRIC^*\left(\mathbf{X}_t,\ \{\mathbf{X}_i\}_{i=t-\tau}^{t-1}\right) &= \mathcal{K}_2^*\left(\left\{\mathbf{X}_t^*\oplus^{\mathcal{D}}\mathbf{X}_i^*\right\}_{i=t-\tau}^{t-1}\right) \\
\text{where } \{\mathbf{X}_i^*\}_{i=t-\tau}^{t} &= \mathcal{K}_1^*\left(\{\mathbf{X}_i\}_{i=t-\tau}^{t}\right).
\end{aligned}
\tag{S6}
$$

Where $\mathcal{K}_1^*(\cdot) : \mathbb{R}^{C\times(\tau+1)\times H\times W} \rightarrow \mathbb{R}^{C\times(\tau+1)\times H_1\times W_1}$ and $\mathcal{K}_2^*(\cdot) : \mathbb{R}^{2C\times\tau\times H_1\times W_1} \rightarrow \mathbb{R}^{C\times\tau\times H_2\times W_2}$ are two 3D convolution blocks with a kernel size of 1 in temporal dimension. Thus, the kernels are temporal-shared for each convolution layer. Then if 2D conv blocks of the same layer in Eq. S5 (the original) are shared, TRIC would be equivalent to its 3D version (TRIC$^*$). In this case, the numbers of parameters of TRIC and TRIC$^*$ are the same, *i.e.*, $(C\times1\times3\times3\times C)+(2C\times1\times3\times3\times C)$.

However, when we review the present implementations of TRIC and TRIC$^*$, we realize that they are all in a stacked form, *i.e.*, $\mathcal{K}_1/\mathcal{K}_1^*$ is always nested in $\mathcal{K}_2/\mathcal{K}_2^*$. It means that, the calculation of $\mathcal{K}_2/\mathcal{K}_2^*$ has to wait for the outcome of $\mathcal{K}_1/\mathcal{K}_1^*$. To this end, a more parallel and elegant implementation of TRIC$^*$ comes up, and Fig. S4 intuitively describes this implementation. The key idea of this implementation is temporal cross reorganization for target-history feature pairs and temporal-shared 3D Conv-based relation encoding with $2\times3\times3$ kernels and time stride of 2. In this case, the

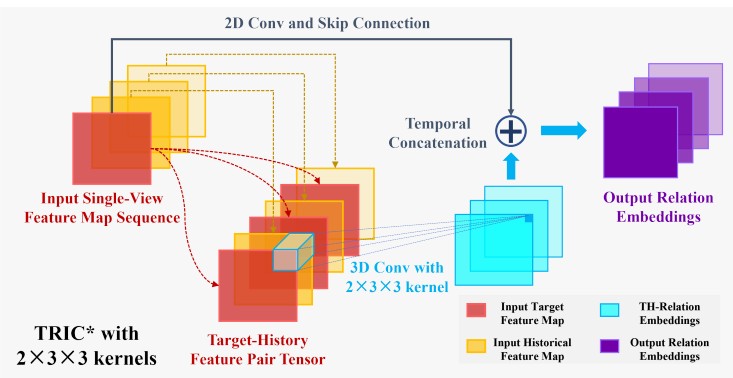

Figure S4: A more parallel implementation of TRIC$^*$.

parameter scale are keeping the same with present TRIC$^*$ with $1 \times 3 \times 3$ kernels, while enjoying a more parallel structure and less computation since there is no depth-concatenation and computation of the two Conv blocks are decoupled. ***More experiments and analysis will be conducted in the future work about these implementations.***

## D    Datasets and training setup

### D.1    Datasets

Three datasets have been used to valid the performance of TARSS-Net, including CARRADA [22], CARRADA-RAC and KuRALs.

**CARRADA** is a camera-radar synchronised dataset with large scale, which contains multi-view annotated radar recordings (RAD tensors) collected from a low-cost FMCW radar in various scenarios under different weathers conditions. There are $4$ categories of objects: *pedestrian*, *cyclist*, *vehicle* and *background*; and multiple same/different types of moving objects could appear in a single time frame. The dimensions of RAD tensor are $256$, $256$ and $64$, respectively. For fair comparisons, RSS-Net and RAMP-CNN were modified as the multi-view version, and all the SoTA methods including our TARSS-Nets were trained with the same setup.

**CARRADA-RAC** is an improved version of CARRADA. As mentioned in §§ 4.1, CARRADA is a camera-radar synchronised dataset. Its RD and RA annotation of were generated in a semi-automatic way [22], in which the unreliable depth estimation of camera image and low angle resolution of FMCW radar reduce the labeling accuracy, especially in RA view. CARRADA-RAC further improves CARRADA by correcting its RA annotation accurate [35]. The training, validation and test subsets were split as in [21].

**KuRALS** is a large-scale self-collected real dataset with refined annotation. It is produced by a Ku-band continuous wave radar and includes various typical targets such as UAVs, airplanes, ships, and cars, which could support several radar sensing tasks including RSS, target tracking and so on. There are only RD tensors in KuRALS, so TMVA-Net, PKCIn-Net as well as TARSS-Net shown in Table 3 are modified to single view version. The complete dataset is forthcoming.

### D.2    Training Setup

The training setup for our proposed models and other compared SoTA networks is strictly consistent as follows:

**Input form**: The raw 3D RAD tensor of each time frame was first compressed as 2D RA, AD and RD views with the input sizes of $256 \times 256$, $256 \times 256$ and $256 \times 64$. For the networks without explicitly considering the temporal information, *e.g.*, RSS-Net [14] and MV-Net [21], 2 historical frames (*i.e.*, $\tau = 2$) were added to form the input sequence. Besides, for the temporal information-based networks, *e.g.*, RAMP-CNN [6], TMVA-Net [21], and our TARSS-Net, more historical frames were used: $\tau = 8$ for RAMP-CNN, and $\tau = 4$ for TMVA-Net and TARSS-Net.

**Running platform**: The experimental platform of this work is RTX 3090 GPU with 24G memory. Both the single-view version and the multi-view version of TARSS-Net can be trained and tested on a single 3090 GPU.

**Hyper-parameters**: All the models were trained with Adam optimizer [15] using the default setting of hyper-parameters, $\beta_1 = 0.9$, $\beta_2 = 0.999$ and $\epsilon = 1e - 8$. The initial learning rate was $1e - 4$, which decayed exponentially with the rate of 0.9 every 20 epochs. The training epochs and mini-batch size were 300 and 6, respectively.

**Evaluation metrics**: For comprehensive comparisons, we evaluated our methods using cell-wise precision/recall, Intersection over Union (IoU), and Dice score. As the principle metrics, mean IoU (mIoU) and mean Dice (mDice) are calculated by category averaging. These two metrics are used to compare the performance of different methods on three validation datasets, as shown in Table 1, Table 2 and Table 3. In ablation experiments, as shown in Table 4, Table 5 and Table 6, precision and recall are also presented to comprehensively measure the effect of each design in TARSS-Net. All the reported results were obtained from the test subset.

# E    Additional experiments

## E.1    Ablation experiment of AD encodingg branch

Since prediction results are generated on RD and RA views, AD view is actually embedded in the decoding branch. To further verify the effect of AD view in the multi-view RSS framework, we conduct an additional experiment on AD encoding branch (AD Enc.), both Baseline-B (replacing TRAM from TARSS-Net with emporal aggregation layers designed in TMVANet) and TARSS-Net_D (TRASS-Net with Depth-TRAP) were used for the experiment. Results in Table S1 clearly show that models can achieve better performance by using additional AD view. This performance improvements mainly come from the information compensation, *i.e.*, AD view compensates RD with angle frequency response in angle domain and compensates RA with Doppler information. Such results also reveal that, taking full advantage of the information in the radar signals is indeed helpful for radar scene understanding. Hence from the perspective of information utilization, our method propose a pathway to joint analysis of multiple spatio-temporal domains of radar data.

Table S1: Ablation on AD Encoding Branch

| Method | RD-View (%) | | RA-View (%) | |
|---|---|---|---|---|
| | mIoU | mDice | mIoU | mDice |
| Baseline-B w/o AD Enc. | 52.8 | 64.3 | 35.5 | 42.4 |
| Baseline-B w/ AD Enc. | 56.1 | 68.0 | 37.7 | 46.2 |
| TARSS-Net_D w/o AD Enc. | 60.1 | 71.9 | 39.8 | 49.2 |
| **TARSS-Net_D w/ AD Enc.** | **63.4** | **75.2** | **41.4** | **51.3** |

## E.2    Comparison of class-wise performances

There are 4 categories of objects in CARRADA dataset, including: pedestrian, cyclist, vehicle and background. The intuitive comparison of class-wise performances is shown in Fig. S5. TARSS-Net_D has the best overall RSS performance for most types of objects, especially for cyclists, while TARSS-Net_S performs better in the segmentation of pedestrians. RSS results are influenced by physical properties and motion states of different objects, which makes Depth-TRAP and Spatio-TRAP play their respective advantages. How to further improve TARSS-Net by combining object characteristics is also one of our future research directions.

## E.3    Features visualization

Some feature visualization of core process in TARSS-Net are shown in Fig. S6. It confirms that the proposed novel temporal information learning paradigm, *i.e.* TRAM, is effective to make network learning pay attention to RSS target.

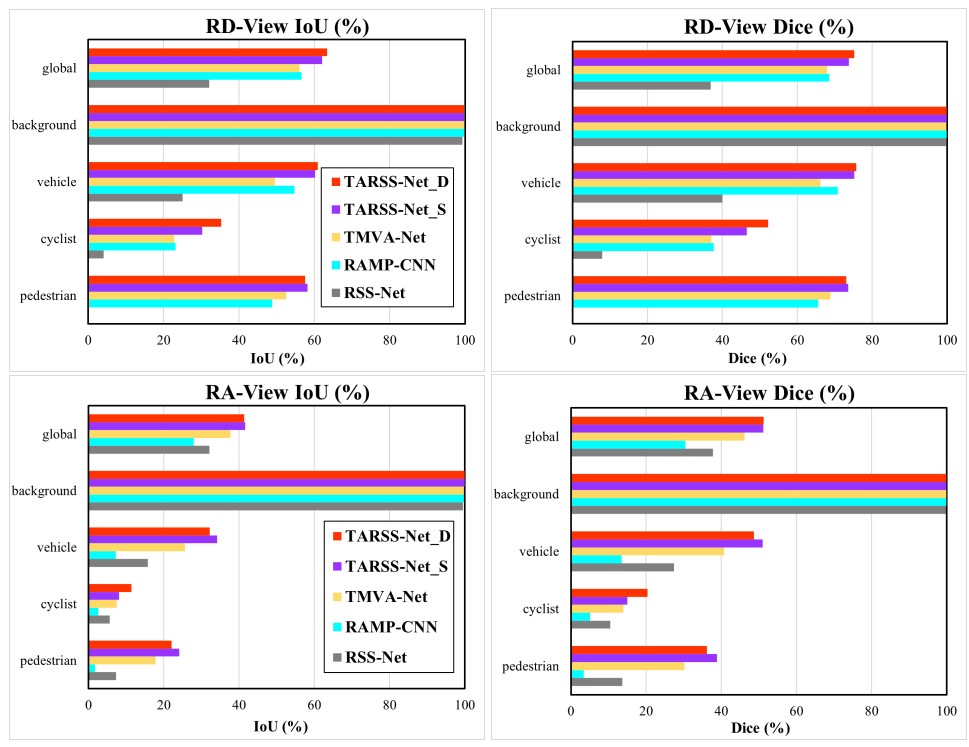

Figure S5: Comparison of class-wise performances.

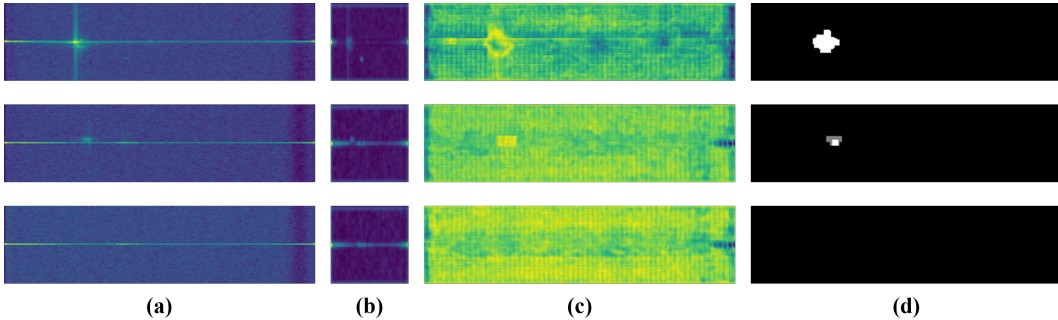

Figure S6: Feature Visualization. (a) Input RD-view frame. (b) The activation response heatmaps of TRAM outputs. (c) TARSS-Net outputs before Softmax. (d) Ground Truth Mask.

## E.4 Visualization of some examples

Some RSS results of TARSS-Net are visualized in Fig. S7. More visualized performances are provided in the supplementary videos.

## F Broader Discussions

**Limitations**. i) As can be seen in Table 1, TARSS-Net performs worse than PKCIn-Net (which incorporates the advantages of classical radar detection theories), T-RODNet and TransRSS (which enhance spatial feature representation based on Transformer) in RA view where the temporal information is not obvious. ii) Deeply exploration on the learning principle and applicable scenarios of TARSS-Net_D and TARSS-Net_S is needed in future work. iii) TARSS-Net makes good trade-off between real-time performance and RSS effects. However, there is still room to improve its inference speed. These limitations also point to the improvement direction of TARSS-Net.

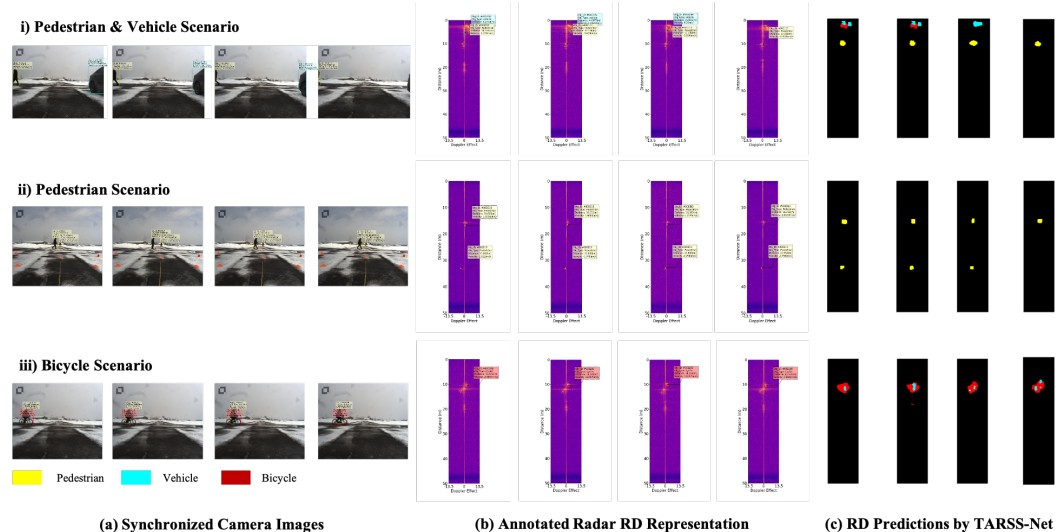

Figure S7: Visualization of some examples.

**Social Impacts**. As for radar detection, TARSS-Net is applicable to various practical applications, such as perceptions for autonomous driving, UAV surveillance and marine monitoring. Furthermore, the proposed novel paradigm of temporal relationship learning is causal and can be exploited by more domains involving sequence data proceeding, including stock prediction, weather prediction, etc. However, inappropriate usage may lead to decreased reliability, potentially resulting in safety and other issues.

