# OpenReview forum: "TARSS-Net: Temporal-Aware Radar Semantic Segmentation Network"
_NeurIPS.cc/2024/Conference — NeurIPS 2024 poster_

### Official Review · Reviewer_kN5s · 2024-07-10

**Soundness:** 3
**Presentation:** 3
**Contribution:** 3
**Rating:** 5
**Confidence:** 5

**Summary:**

This paper proposes a novel framework designed to integrate temporal information into radar-based semantic segmentation tasks. To achieve more effective temporal information integration, the framework introduces two key modules: the Target-History Temporal Relation Encoder (TH-TRE), which analyzes the relationships between different time frames, and the Temporal Relation Attentive Pooling (TRAP) module, which aggregates information along the temporal axis.

**Strengths:**

This paper introduces a novel algorithm for radar-based semantic segmentation tasks. The manuscript is well-constructed, with the discussion section clearly articulating and comparing the differences between various algorithms. The newly designed algorithm demonstrates a notable improvement in performance without increasing the model size.

**Weaknesses:**

Please consult the question section for further information.

**Questions:**

1) Could the authors provide a comparison in terms of computational time?

2) Although the primary baseline, PKCIn-Net, also incorporates temporal information, the difference in performance between the proposed algorithm and PKCIn-Net appears to be minimal.

3) Additionally, the authors mention that this paper considers compression from both spatial-temporal and depth-temporal dimensions. Therefore, a pertinent question arises: would it be possible to first fuse spatial and depth information, and then subsequently fuse this combined data along the temporal axis?

**Limitations:**

Please consult the question section for further information.

---

> ### Author Rebuttal · Authors · 2024-08-07
>
> # **Questions**
>
> **[Q1. COMPUTATIONAL TIME COMPARISON.]** There is real-time performance comparison **in `Sec. E.2 of Appendix`**, which shows the comparison between TARSS-Net and some other models, involving**model size**, **calculation amount**, **real-time performance** and other information of in details. In addition, we also add the VIT model in`Table S2` as a baseline for comparison, which can also be found in`Table 1 of attached PDF` for this rebuttal.
>
> **[Q2. MINIMAL DIFFERENCE IN PERFORMANCE COMPARED WITH PKCIN-NET.]** PKCIn-net makes the classical radar target detection CFAR principle deeply learnable, and proposes a radar-specific target detection operator PeakConv (PKC). **The focus of PKC is not on incorporating temporal information, but we include it in the comparison as it is a landmark work on introducing deep learning to RSS.** Under the premise of the same model size, TARSS-Net outperforms PKCIn-Net by an average of 2% (on all data-view and three dataset). First of all, given the particularity and difficulty of RSS tasks, **a 2% performance improvement is significant**, we also illustrate in our responses to R. MgaH how the RSS performance has improved little by little (if you want to learn more, please see response of `W2 to R. MgaH`). Secondly, PKCIn-Net focuses on making the classical radar detection theory learnable, and TARSS-Net focuses on introducing an efficient temporal relationship modeling method suitable for radar signals. In the later work, **we will also consider adapting PKC to TARSS-Net to further improve the performance of the model**, since TRAM and PKC are not in conflict due to their different focuses. Both PKCIN-Net and TARSS-Net point the way for the development of the RSS field. That is, compared with directly applying the mature learning method in other field (such as CV, NLP, etc.), designing the learning paradigm suitable for the characteristics of radar signals is more beneficial.
>
> **[Q3. WOULD IT BE POSSIBLE TO FIRST FUSE SPATIAL AND DEPTH INFORMATION, AND THEN SUBSEQUENTLY FUSE THIS COMBINED DATA ALONG THE TEMPORAL AXIS?]** This is a particularly valuable question. When we originally designed the TRAP module to implement temporal relation measurement and weighted-fusion, we intended to incorporate spatio-temporal information at the same time. However, for the space and depth with high dimensions, it is the most economical choice to do dimensionality reduction and fusion processing separately, i.e., Spatio/Depth-TRAP. **If we want to fuse the two dimensions at the same time, it is bound to introduce modules with larger parameters, so we did not explore it in this work.** Therefore, in the subsequent network optimization process, **considering the complexity** of RSS network to facilitate its application, **we form two versions of Spatio-TRAP and Depth-TRAP on the premise of ensuring their performance advantages**. Your idea is very meaningful, and as mentioned in the `limitations Section in Appendix F (L1017)`, deeply exploration on the learning principle and applicable scenarios of TARSS-Net_D and TARSS-Net_S is needed in future work. Based on this, we look forward to working out an exciting solution to your pertinent question.

---

> > ### Author Response · Authors · 2024-08-14
> > **We appreciate your respect for our hard work**
> >
> > Dear Reviewer kN5s,
> >
> > We have spent a significant amount of time and effort analyzing your concerns and suggestions, and have provided detailed explanations and corresponding modifications. We believe that our response should be sufficient to address your concerns. We **sincerely hope that you would take the time to read our response**. If our response is **adequate**, we kindly ask you to give a **fair score upgrade**; if you still **have other concerns**,  looking forward to **further discussions with you**. Thank you for your contribution to improving the quality of our paper, and we also appreciate your respect for our hard work.
> >
> > 9831 Authors

---

> ### Author Response · Authors · 2024-08-13
> **Looking forward to your comments on our reply**
>
> We have provided detailed responses to all of your questions. Hope to get your approval on the reply and update the rating. We also look forward to more in-depth communication and discussion with you. Thank you again!

---

### Official Review · Reviewer_WAmw · 2024-07-11

**Soundness:** 3
**Presentation:** 3
**Contribution:** 2
**Rating:** 5
**Confidence:** 4

**Summary:**

In this work, the authors created a network called TARSS-Net for radar semantic segmentation. Compared to traditional methods, the authors emphasized the superiority of their approach in the clever utilization of historical information. Specifically, TARSS-Net incorporates a module called TRAM, which is designed similarly to the attention mechanism. This module first learns the relationship between historical frames and the current frame through the TH-TRE module, and then maps this learned relationship to hidden representations via TRAP. TRAM is specifically designed for radar semantic segmentation, ensuring that computational complexity remains manageable while achieving efficient and accurate radar semantic segmentation across three real radar segmentation datasets.

**Strengths:**

The main advantages of the paper are as follows:

(1) The paper provides a comprehensive review and analysis of the application background, existing methods, and challenges of RSS semantic segmentation.

(2) The authors have significantly enhanced the model performance in RSS semantic segmentation tasks by balancing accuracy and efficiency through carefully designed methods, which holds practical significance.

**Weaknesses:**

629/5000
This paper may have the following potential shortcomings:

(1) Lack of Clear Hypotheses and Reasoning: The authors designed a novel temporal modeling paradigm for RSS and claimed its effectiveness in the context of RSS semantic segmentation. However, they seem to provide their viewpoint directly without offering clear hypotheses and reasoning about why this paradigm is effective for RSS semantic segmentation, which may confuse the readers.

(2) Insufficient Emphasis on the Design Motivation: The authors lack an emphasis on the motivation behind the model design. For instance, regarding the design of the TH-TRE module, the authors aim to use this module to encourage the model to focus more on high-dimensional information of the current time frame. However, Section 3.1 lacks an emphasis on this design motivation and focuses too much on formula statements. Given the considerable complexity of the designed method, it is crucial to introduce the design concept behind the module to the readers.

(3) Issues with Paper Formatting and Visualization: There are some issues with the paper's formatting and visualization. For example, the sequence of Figures 4 and 3 does not align well with typical reading habits, leading to some reading difficulty. Additionally, some elements in the figures are not drawn rigorously; for instance, the input of the TH-TRE module should be a 4D tensor, but Figure 3 depicts it more like a 3D tensor. For readers accustomed to visual aids, the authors' lack of rigorous visualization might lead to a misunderstanding of the model details.

(4) Potential Issues in the Experimental Section: There may be shortcomings in the experimental section, especially in comparison with the use of the Self Attention (SA) model in the temporal domain. Although the authors mention that applying SA in the temporal domain might lead to excessive computational consumption, the appendix shows that they have implemented a baseline model applying SA in the temporal domain based on ViT. Therefore, a more rigorous experiment should include this baseline model in the accuracy analysis (since accuracy analysis does not involve computational efficiency), as the authors have criticized this method in their analysis.

**Questions:**

Some questions are as follows:

(1) What is the core innovation of this paper? Causal dilated convolutions also seem to align with the paradigm proposed by the authors. Compared to these simpler methods, what advantages does the method in this paper have? Why is it more advantageous?

(2) Is it meaningful to discuss real-time performance for RSS semantic segmentation? If RSS semantic segmentation in most scenarios demands higher accuracy rather than efficiency, then the criticism of other methods in terms of efficiency in the paper seems less significant. Can the authors provide specific examples of scenarios to analyze and explain this?

**Limitations:**

The authors adequately discuss the limitations.

---

> ### Author Rebuttal · Authors · 2024-08-07
>
> ## Weaknesses
>
> **[W1. LACK OF CLEAR HYPOTHESES AND REASONING.]** Sorry for the confusing, and you are right that clear hypotheses and reasoning are important. However, at present, the data-driven learning training of AI models allows researchers to conceptuate and design algorithms at a higher level (the functional and motivated level), which means the classification interface (Modeling functions of network) is no longer expressed in an explicit way by formulas, but is learned implicitly in the network parameters. It solves the problem that the classification interface cannot be shown by existing expressions. In this way, the algorithm performance has been further improved and generalized. **The writing of TARSS-Net exactly follows this theoretical system of priori interpretability.** Of course, **TARSS-Net essentially involves a series of theories and derivations of feature engineering, metric learning, differentiable layer design** and so on with radar signal as input. Due to the space limitation and ease of understanding for readers with general AI research background, this paper starts from the high level design and implementation, but the necessary derivations in key parts are preserved.
>
> In addition, this paper gives a detailed introduction on why TARSS-Net  is effective for RSS. We have conducted a comprehensive discussion for existing time series modeling paradigms, and analyze their own drawbacks and the factors that make them unsuitable for RSS (see `Sec. 2`). Based on the above analysis, we illustrate the design motivation of TARSS-Net for RSS task one by one (see `L144-L155`), as well as detailed implementation methods in `Sec. 3`. Also, we further verify the superiority of TARSS-Net over the existing methods that consider temporal relation information.
>
> We believe that **the confusion you mentioned can be eliminated after readers carefully read the full paper, the Appendix and the code**.
>
> **[W2. INSUFFICIENT EMPHASIS ON THE DESIGN MOTIVATION.]** In `Sec. 2`, we have conducted **a comprehensive discussion for existing temporal modeling paradigms**, and based on the above analysis, we illustrate **the design motivation of TARSS-Net for RSS one by one**, including the design motivation of TH-TRE module. We believe that after you go back and read the first two sections of this paper again, your questions will be answered.
>
> **[W3. ISSUES WITH PAPER FORMATTING AND VISUALIZATION.]** Sorry for the reading trouble caused by unreasonable formatting and visualization. Due to the space limit of paper submission, we had to make some typography which might make it uncomfortable to read. **These will be corrected in next manuscript version, including the order of Fig. 3 and Fig. 4, more rigorously draw for elements in the figures, etc**.
>
> **[W4. POTENTIAL ISSUES IN THE EXPERIMENTAL SECTION.]** Due to space limitation, we show the experimental results that can best help to verify the performance of TARSS-Net in the most concise way. **Due to the sparsity of radar targets, the dense computation of SA will inevitably introduce redundant computations on irrelevant information thus degrading the RSS performance**. The performance of the VIT model is supplemented in `Table 1 of attached PDF` for this rebuttal. We also promise to **add it to Table S2 in the Appendix** of the revised manuscript.
>
> ## Questions
>
> **[Q1. THE CORE INNOVATION OF THIS PAPER.]** The core innovation is to propose an effective temporal modeling method specific to RSS tasks, i.e., the **plug-and-play TRAM which combines the advantages of causality, end-to-end learnability, constant model parameters under arbitrary length input, and linear growth of computational complexity with the length of the sequence**. These advantages cannot be satisfied in the same time when using other existing temporal modeling methods including Tranformers, 3DConv, RNNs and HMMs. For its significance, innovation and advantages, please read the first two Sections of the paper in details. In terms of **causal dilated convolution (CDC)**, it definitly meets the parallel-computation, larger RF with fixed-size kernels and causal computing mechanism, however **the dilation rate should be pre-defined**, i.e., if the input length changes, the hyper-parameter, dilation rate should be changed accordingly before training. While **TRAM does not require any adjustment when handeling different lengths of input**. Morover, as far as we know, **CDC is not in the 3D form which has the limitation for handeling temporal-spatial data** such as radar RAD sequence. Hence, instead of talking about CDC let's dive into 3DConvs, which are more prefered by the researchers in RSS field.
>
> **[Q2. IS IT MEANINGFUL TO DISCUSS REAL-TIME PERFORMANCE FOR RSS?]** Yes, it is very important to discuss the real-time capability of RSS. As a remote sensing device, radar is applied in many fileds, such as automatic driving, security warning and so on. Taking Ku-band drone surveillance radar as example, the PRT (pulse repetition time) is around 80us, and it has 128 coherent pulses in one CPI (one Range-Doppler frame), then the data rate for detection will be $\frac{1 \times 10^6}{80 \times 128} = 97.66~\text{FPS}$. This requires subsequent signal processing and detection/segmentation algorithms to match this data rate as much as possible. Hence, in order **to accurately detect and stably track the target, real-time performance is one of the important indicators of RSS task**, which has practical significance at the application level. Taking automatic driving as another example, the moving car needs real-time feedback of detection results in the surrounding environment, otherwise it will lead to unexpected consequences. Therefore, RSS needs to balance accuracy and efficiency.

---

> > ### Comment · Reviewer_WAmw · 2024-08-12
> >
> > Thank you for your response and the detailed explanation of my questions. I believe addressing these issues is crucial for refining TARSS-Net. Regarding the authors' response to [W1. LACK OF CLEAR HYPOTHESES AND REASONING], as a reader, it is beneficial to understand the characteristics of the input data in specific scenarios and the challenges associated with processing it. A clear thought process—such as the analysis and reasoning behind the hypotheses proposed by the authors, followed by the methods tailored to specific scenarios—can help readers with a general AI research background better comprehend the intentions behind the authors' work. In this regard, one of the baseline methods in this paper, TransRSS, presents its ideas more effectively in the Introduction section.

---

> > > ### Author Response · Authors · 2024-08-13
> > > **Further response to [W1]**
> > >
> > > Dear Reviewer WAmw,
> > > We appreciate your recognition of our response and further suggestion for modification. TransRSS indeed sets a good example in the radar detection background introduction, with a clear chain of thought. However, we did not choose this writing logic for the following two reasons:
> > >
> > > 1. **TARSS-Net has different methodology** from that of TransRSS, the essence of the problem in TARSS-Net's hand is the efficient temporal encoding of radar high-dimensional spatio-temporal tensors. In the initial version of TARSS-Net's introduction, we also began by *related background knowledge*, including the similarities and differences between radar devices and other visual devices, radar signal processing pipeline, and the characteristics of the radar data in hand. Then, we summarized *the development of radar target detection methods along a timeline*, ultimately highlighting the importance of temporal relationship modeling. However, unlike TransRSS, which combines Transformer and CNN in a way that is more easily understood and accepted by general AI researchers, such writing style would fail to allow readers to realize the limitations and challenges of existing temporal modeling methods in radar signal processing. That is, **given the existence of methods like 3DConv, RNNs, and Transformer, why is there a fundamental need for TARSS-Net?** After repeated discussions and revisions, we chose to quickly **highlight the current state of research in temporal relationship modeling in RSS** in the limited space of the Introduction, **point out the research gaps**, and then provide **a detailed summary of existing temporal modeling methods** in Sec. 2. We deeply analyze why existing methods are not suitable for RSS and **how to design temporal modeling methods suitable for the RSS field** by addressing these limitations.
> > > 2. Currently, **TARSS-Net has a higher research starting point**. While organizing this work, we found that excellent works like TMVA, PKC, and TransRSS *already possess the complete chain of thought you expect in their Introductions*. Therefore, we boldly omitted some common background knowledge that has been thoroughly discussed, allowing us to *take a higher perspective to discuss and analyze the RSS task from the entry point of temporal modeling mechanisms*. This represents new cognition and understanding not present in current research work in this field. We believe these contents can bring new insights and more inspiration to readers, which is also why we believe this work is suitable for the NeurIPS community. TARSS-Net, standing on the foundation of existing excellent work, brings fresh perspectives and cognition.
> > >
> > > However, we must admit that since we placed more emphasis on temporal modeling, we had to omit some background information that does not affect the understanding of this paper under the constraints of the current limited space. This background information is included in the appendix. For example, in Sec. A From Radar Signals to Multi-View Representations, readers can understand the detailed process of millimeter-wave radar data processing in autonomous driving scenarios. In Sec. C Additional Descriptions of TARSS-Net, readers can gain a deeper understanding of the overall thought process of TARSS-Net in conjunction with the Methodology described in the manuscript. However, we agree with your suggestion. **Since additional page will be allowed in the camera-ready version, we promise to supplement the background introduction that cannot be added at this stage in the introduction, including the *characteristics of radar data* and *the challenges associated with processing it* you mentioned**, to better present the ***clear thought process*** of this paper. This will allow readers to easily grasp our intentions without having to read other materials or appendix as much as possible. Thank you for your contributions and thoughts to improve the quality of this paper. We sincerely express our respect to you and if you are satisfied with our new response to **[W1]**, we also look forward to your higher evaluation, so that more readers can see the higher quality TARSS-Net after the revision!

---

> > > ### Author Response · Authors · 2024-08-14
> > > **Please take some time to review our new responses**
> > >
> > > Dear Review WAmw,
> > >
> > > Thank you for you hard work! The discussions we have had not only help to improve the quality of this paper but also show respect for our hard work. Thank you again. We have responded to your new concerns accordingly. Since the rebuttal is almost over, please take some time to review our new responses. **If you are satisfied with the replies, we especially appreciate you raising the evaluation score**. If there are any **new concerns**, we look forward to continuing the **in-depth discussion with you**.
> > >
> > > Sincerely,
> > >
> > > 9831 Authors.

---

### Official Review · Reviewer_awi8 · 2024-07-12

**Soundness:** 3
**Presentation:** 3
**Contribution:** 3
**Rating:** 5
**Confidence:** 4

**Summary:**

This paper primarily introduces a network model called TARSS-Net, designed for the task of radar semantic segmentation. It effectively utilizes the temporal information of radar signals by introducing a novel temporal modeling mechanism, enhancing radar semantic segmentation performance. Specifically, the paper proposes a module called TRAM for temporal relationship learning between the target and historical frames, integrating it with other core components to construct the TARSS-Net model.

**Strengths:**

1.The author successfully applies time modeling to the radar semantic segmentation task, achieving state-of-the-art performance on the RD-View radar semantic segmentation benchmark.
2.Rigorous ablation studies were conducted, providing solid evidence of the proposed method's efficacy.
3.The paper provides a good review of time modeling, which is helpful to the radar semantic segmentation task.

**Weaknesses:**

1.This paper applies the time modeling to the field of radar semantic segmentation reasonably, but there are few innovations in the paper.
2.The performance of time modeling approaches tends to degrade in crowded scenarios.
3.It is difficult to significantly improve the performance of RA-View under the time modeling method
4.The related works reviewed for the radar semantic segmentation task are not comprehensive.
5.The placement of illustrations within the main text is not particularly reasonable, making re-display in the Appendix an unsatisfactory choice.

**Questions:**

1.Can you provide a more detailed analysis of the computational complexity?

---

> ### Author Rebuttal · Authors · 2024-08-07
>
> ## Weaknesses
> **[W1. FEW INNOVATIONS.]** Thank you for recognizing the rationality of our approach. However, beyond reasonability, the novelty of TARSS-Net is also guaranteed. We feel sorry that we failed to make you see the novelty, as well as the careful thought and extensive experimental validation that went into it. Therefore, we think it is important to reiterate the real value and advantages of TARRS-Net. As we covered in `Sec. 2`, if you want to exploit temporal information in radar data, then you basically have 4 options: HMMs, RNNs, 3DConvs and Transformers. However, when we decide to use them, we encounter some unavoidable problems: i) **as shallow probabilistic models, HMMs are very limited in their ability to express primitives** for each single time step, and they **cannot be trained end-to-end** (`L86~L89`); ii) **RNNs** inherit the causal computing mechanism of HMMs and can be deeply trained end-to-end, but they **cannot fully enjoy the computational efficiency of parallelization** (`L90~L100`); iii) **3DConv** can be fully parallelized, but its calculation nature is non-causal and cannot be adaptively adjusted according to the length of the input sequence, i.e., **to process longer sequences, the network needs go wider or deeper**, which leads to the introduction of more training parameters (`L102~L114`); iv) the **Transformer** with SA as the core overcomes the problem of limited local receptive field when 3DConv deals with long sequences, and also overcomes the defect of RNNs that cannot be parallelized. However, its **computational complexity will show square level growth with the length of the sequence, resulting in computational redundancy** (`L115~L127`). Seeing these problems, we deeply realized that there is **NO FREE LUNCH**, we have to **redesign temporal modeling method for radar data, and RSS model should not be stagnant on these methods, it should have better sequence modeling methods**. To this end, we meticulously redesign the temporal modeling method and propose the design principles, resulting in **TRAM**, which **combines the advantages of causality, end-to-end learnability, constant model parameters under arbitrary length input, and linear growth of computational complexity with the length of the sequence**. Without careful design, we will not find a model in current methods that can achieve all of these advantages and still have the performance of SoTA. Therefore, we hope you to understand our efforts and recognize our work.
>
> **[W2. PERFORMANCE IN CROWDED SCENARIOS.]** The concern you raised is indeed worth discussing. In real-measured radar data, **target signatures often show extremely sparse characteristics** (this does not refer to the physical size of the target, but to the characteristics of the target reflected in the radar signal)，so **sparse and small target detection is a classic pain point which radar signal researcher** dedicated to solving. So, if we understand your concern correctly, the RSS task does not suffer from the problem of model performance tends to degrade in crowded scenarios, which is one of the reasons why TARSS-Net **emphasize the importance of learning and exploiting temporal relationships to enhance the representation of sparse target signature** for RSS.
>
> **[W3. DIFFICULTY OF PERFORMANCE IMPROVEMENT ON RA-VIEW.]** This is a good question that points out something unique to the RSS domain. The mismatch between the quality of RA-view data (for range and angle measurement) and RD-view data (for range and Doppler measurement) is a common phenomenon. Restricted by the hardware, most radars pay more attention to the ranging accuracy when designing, and the angular accuracy of the commonly used low-cost radar is hard to guarantee. Therefore, the difficulty in improving the segmentation accuracy of RA is more caused by the poor data quality, rather than the failure of the temporal modeling method. [35] notices this problem and modifies annotation quality of RA, resulting CARRADA-RAC dataset. Comparing `Table 1 and Table 2`, it can be seen that annotation correction can improve the performance of RA-view from 51.3% to 58.7 % (TARSS-Net_D). Therefore, it is reasonable to think that there is more room for the temporal modeling method to play a role after further improving the data quality.
>
> **[W4. NOT COMPREHENSIVE RELATED RSS WORKS.]** Temporal modeling is not an emerging topic in other fields. However, **effective and necessary modeling of temporal relationships has not been fully emphasized in RSS**. Therefore, in order to encourage readers to revisit the temporal relation modeling from the perspective of radar signal processing and understand the motivation of TARSS-Net, **we focus on the analysis of existing temporal modeling methods, the obstacles to their application in RSS domain, and the elaborate design of TARSSnet to face those obstacles**. It is believed that `Sec. 1 and 2` can bring somehow inspiration for readers in the NIPS community. It is worth noting that **in SoTA method comparison (see `Sec. 4.2`), we have included as comprehensive as possible the existing excellent RSS works** and made a brief analysis. Limited by the length of this paper, please visit the relevant index to download original paper for details of these works.
>
> **[W5. PLACEMENT OF ILLUSTRATIONS.]** Thanks for your kind reminder and we are sorry for the reading trouble caused by unreasonable placement. Due to the space limit of paper submission, we had to make some typography which might make it uncomfortable to read. **These will be corrected in next manuscript version, including the transposition of the order of Fig. 3 and Fig. 4, as far as possible to supplement the main text to reduce the redisplay in Appendix, etc**.
>
> ## Questions
> **[Q1. MORE DETAILED ANALYSIS OF THE COMPUTATIONAL COMPLEXITY.]** There is real-time performance comparison in `Sec. E.2 of Appendix` and `Table 1 of PDF for rebuttal`, including model size, MACs, FPS and metrics.

---

> > ### Comment · Reviewer_awi8 · 2024-08-12
> >
> > Thank you for your response. I appreciate your further explanation of the TARRS-Net and the additional analysis of computational complexity. It is noteworthy that Crowded Scenarios refer to situations with a higher number of objects, and typically, time modeling may experience performance degradation in such scenarios.
> >
> > Overall, the authors have addressed some of the issues raised during the review process, reinforcing the significance of TARSS-Net as an innovative approach in the field of radar signal processing. I look forward to seeing the proposed improvements, and I will maintain my current rating.

---

> > > ### Author Response · Authors · 2024-08-12
> > > **Replying to Official Comment by Reviewer awi8**
> > >
> > > First, we want to thank you for recognizing the contributions and innovations of our paper. Then thanks for your more detailed explanation about **[W2 PERFORMANCE DEGRADES IN CROWDED SCENARIOS]**. Your mention of *multiple targets* has clarified your definition of *crowded scenarios*. The common multi-target situations in radars are already included in the CARRADA and our self-collected KuRALS datasets (e.g., the simultaneous presence of pedestrians and cars, multiple drones/UAVs or multiple ships). For the crowded scenarios you mentioned, we believe the closest examples are **drone swarm detection** and **bird flock detection**:
> > >
> > > - Taking the **drone swarm detection** scenario as an example. Typically, the radar will first perform a wide-area search, i.e., in the scanning mode it will emit coherent/incoherent pulse trains in each direction. The use of coherent pulse trains is more common. In each direction, these coherent pulse trains will form what we call RD representations. Considering the flight safety distance of the drones themselves, **the number of targets that can be covered in the same direction (i.e., one RD frame) will be much smaller than the entire swarm size**. In the end, it is difficult for single RD frame to reflect the crowded scenarios that appear in imaging signals such as camera images and SAR data. Subsequently, radar gets in the tracking mode, the system will form multiple tracking channels with smaller detection range, **making each channel cover only the spatial range of one target as much as possible**. After accumulating tracks over multiple RD frames, it will perform target identification or other more refined perception tasks. Therefore, whether in the scanning or the tracking mode, **it is difficult to obtain RD frame which can reflect so called crowded scenario**. Moreover, collecting such radar data for drone swarms is very challenging, requiring the cooperation of very professional flight control technicians and radar technicians to successfully collect this type of data.
> > > - In the **bird flock detection** scenario, the flight distance of birds (about 1~2 meters during the migration of large birds and tens of centimeters when small birds are foraging) is often lower than the radar's range resolution. For example, the Ku-band radar we use for collecting KuRALS dataset has a radial range resolution of about 3 meters. Therefore, **the bird flock targets will all appear as a single mass target on one RD frame**. At this point, *the crowded scenario in vision collapses into a connected domain in the radar perspective*.
> > > - Finally, let's analyze a more common crowded case, namely **heavy traffic**. Unlike vision, the form of crowded traffic scenarios in RDs cannot satisfy what you mentioned as crowded scenarios. This is because the R axis means radial distance, and D axis means Doppler or velocity. In heavy traffic, the speeds of the moving targets are similar, which results in a line pattern in parallel with R axis of RD frame. Such situations are included in the dataset involved in our experiments, where TARSS-Net can demonstrate its effectiveness.
> > >
> > > In summary, the *crowded scenarios* you refered to may appear more frequently in camera or SAR imaging radar. However, in the pulse Doppler or continuous wave radar discussed in this paper, crowded scenarios are generally diluted across different channels at signal processing level or collapsed into a single "target" due to the radar's physical characteristics. Therefore, **visual crowding cannot be directly mapped to the radar scenarios discussed in this paper**. The crowded scenarios you mentioned exceed the scope of signal processing algorithms and are a systematic issue. Radar designers will consider multiple aspects to design corresponding solutions, not just relying on signal processing algorithms.
> > >
> > > TARSS-Net proposes a general method for improving RSS performance. The datasets used in this paper cover typical radar application fields, such as autonomous driving and low-altitude surveillance. The experimental results also demonstrate the effectiveness of TARSS-Net. Hence the impact of crowded scenarios on the performance of radar temporal modeling methods is not within the scope of this paper. Nonetheless, thanks for your valuable concern, which motivates us to do more inspiring discussions and points us in a new research direction of this field. We will specifically consider this situation in subsequent studies. The TARSS-Net proposed in this paper has already demonstrated its effectiveness in improving the performance of general radar detection scenarios. We are pleased that this has been recognized by you and other reviewers. If our response can address your concerns, we hope you can further provide a higher rating to inspire more peers and further improve radar signal processing capabilities.

---

> > > > ### Author Response · Authors · 2024-08-13
> > > > **Looking forward to your comments on our new reply!**
> > > >
> > > > We have updated responses to your new comments. Hope to get your approval on the reply and update the rating. We also look forward to more in-depth communication and discussion with you. Thank you again!

---

> > > > ### Author Response · Authors · 2024-08-14
> > > > **Please take some time to review our new responses**
> > > >
> > > > Dear Review awi8,
> > > >
> > > > Thank you for you hard work! The discussions we have had not only help to improve the quality of this paper but also show respect for our hard work. Thank you again. We have responded to your new concerns accordingly. Since the rebuttal is almost over, please take some time to review our new responses. **If you are satisfied with the replies, we especially appreciate you raising the evaluation score**. If there are any **new concerns**, we look forward to continuing the **in-depth discussion with you**.
> > > >
> > > > Sincerely,
> > > >
> > > > 9831 Authors.

---

### Official Review · Reviewer_MgaH · 2024-07-16

**Soundness:** 2
**Presentation:** 3
**Contribution:** 2
**Rating:** 4
**Confidence:** 4

**Summary:**

This paper proposed a temporal-aware framework, TARSS-Net, to enhance Radar semantic segmentation. The key idea is to propose a Temporal relation attentive module, TRAM (consists of Target-History Temporal Relation Encoding [TH-TRE] and Temporal Relation-Aware Pooling [TRAP] ), to capture the relations between Radar sequences.  Comparisons with baseline methods on three datasets (CARRADA, CARRADA-RAC and a self-collected KuRALS) show the advantages of the proposed method

**Strengths:**

- The paper is well-written and in details. The motivation of incorporating temporal information is practical.
- The proposed method outperforms the SOTA methods in some aspects, especially in RD-View.
- The experiments and ablation studies are extensive.

**Weaknesses:**

-   Since there are many existing works to incorporate spatio-temporal information, the technical contribution is limited without the insights considering the data characteristic of Range-Angle-Doppler data.
- The performance increment is marginal, although various modules are designed.
- It lacks of runtime statics to evaluate the time consumption introduced by temporal design.

**Questions:**

- How are the results when two frames are considered?
- The performance decreases after 6 frames and increases again from 9 frames. Is there any analysis of that?
- Can the authors provide to some feature visualization to confirm that whether the designed scheme payed attention to the object (or where to pay attention) in Radar sequences.

**Limitations:**

The authors appropriately discussed the limitations in the appendix.

---

> ### Author Rebuttal · Authors · 2024-08-07
>
> ## Weaknesses
>
> **[W1. LIMITED TECNICAL CONTRIBUTION.]** Sorry to have caused the reviewer such concerns. The core point of this paper is to **redesign a better spatio-temporal modeling method for radar data from the perspective of temporal information utilization**. Indeed, there are many works in other fields that discuss the utilization of spatio-temporal information. However, as we analyzed in `L34~L44`, **there is still a large research gap in the related research of radar-oriented deep learning models**: "In terms of temporal information utilization, the common practice is still 3DConv." Hence, we dedicated to exploring an advanced temporal modeling mechanism, whcih is urgently needed for modern radar (alleviating the problem of time-sensitive changes in radar data quality and learning better target representations). For the insights of RAD data handling, it is not the main focus of this paper, for two reasons: i) **TMVA-Net [21] had provided a multi-view learning approach that well balances information utilization and efficient processing for RAD data**; ii) **conducting detection/segmentation on RAD data is not universal in radar systems**, for some phased array radar system, it is more efficient to perform the detection directly on RD, e.g., KuRALS dataset used in this paper. In summary, this paper aims at the problem of temporal modeling which is more general in radar systems. From this perspective, this work summarizes the advantages and disadvantages of modern temporal modeling methods in `L82~L127` and the problems that need special attention when processing radar data in `L130~140`. We will reiterate our core research objectives in the Introduction section to highlight our contributions.
>
> **[W2. MARGINAL PERFORMANCE INCREMENT.]** Sorry not to surprise you, but **this is actually not a small improvement for RSS tasks**. As we mentioned in `L20~L24`, radar data is different from image data. It is susceptible to various interference and does not have semantic information, which make RSS more chanllenging. From the experimental part, it can be seen that the general semantic segmentation model of CV has not achieved good results on radar data (such as FCN, Unet). Then TMVA-Net [21], has improved the performance of RSS by about 2% (taking RD view as an example). The performance improvement of subsequent TransRadar [5], TransRSS [36] and PKCIn-Net [35] also increases by about 2%. To date, from FCN to TARSS-Net, RSS performance has improved from 66% to 75% on benchmark dataset, CARRADA. As you can see, **RSS performance is thus advanced bit by bit**. That said, **the authentic and reproducible 2% performance improvement** achieved by TARSS-Net is significant for the progress in RSS field.
>
> **[W3. LACK OF RUNTIME STATISTICS.]** Due to page limitation, the runtime performance comparison is listed in `Sec. E2 of Appendix`. TARSS-Net_D takes 43ms to infer one RAD@5Frames, 9 ms for one RD@5Frames and 9 ms for RA@5Frames (tested on single Nvidia3090).
> ## Questions
>
> **[Q1. RESULTS WITH 2 FRAMES.]** Thanks for your suggestions to make our experiment more comprehensive and solid. We further tested TARSS-Net_D with 2 input frames on RD view CARRADA: mDice 73.7%, mIoU 62.0%, Precision 71.3%, Recall 76.5%. It is supplymented in `Figure 1 of attached PDF` for this rebuttal.
>
> **[Q2. ANALYSIS OF PERFORMANCE DECREASING AFTER 6 FRAMES AND INCREASING AGAIN FROM 9 FRAMES.]** This is a particularly valuable question. We did do some analysis of the trend of the curve shown in `Fig. 5`, and try to summarize some general conclusions that can guide TARSS-Net usage. Different operating conditions of radar affect the quality of each data frame, so in one RSS dataset, **it is hard to generalize how many historical frames are closely related to the target one**, or which historical frames are helpful to the current frame (poor quality history frame will introduce a lot of irrelevant noise and lead to performance degradation). Not to mention different RSS datasets. All networks dealing with temporal-sturctured radar data face the problem that they cannot calculate the correspondence between the number of input frames and network performance. **TARSS-Net has its own advantages to help ease the choice of input frame length**. With the design of TRIC layer, TARSS-Net could accept input with adjustable time length while keeping the number of parameters constant. When using it, readers can try and choose the number of input frames according to constraints of inference performance of the hardware (TFLOPS) without worrying about model size.
>
> **[Q3. Feature visualization.]** Please see `Figure 2 in attached PDF` for this rebuttal.

---

> > ### Author Response · Authors · 2024-08-14
> > **We appreciate your respect for our hard work**
> >
> > Dear Reviewer MgaH,
> > We have spent a significant amount of time and effort analyzing your concerns and suggestions, and have provided detailed explanations and corresponding modifications. We believe that our response should be sufficient to address your concerns. We **sincerely hope that you would take the time to read our response**. If our response is **adequate**, we kindly ask you to give a **fair score upgrade**; if you still **have other concerns**,  looking forward to **further discussions with you**. Thank you for your contribution to improving the quality of our paper, and we also appreciate your respect for our hard work.
> >
> > 9831 Authors

---

> ### Author Response · Authors · 2024-08-13
> **Looking forward to your comments on our reply**
>
> We have provided detailed responses to all of your questions. Hope to get your approval on the reply and update the rating. We also look forward to more in-depth communication and discussion with you! Thank you again!

---

### Author Rebuttal · Authors · 2024-08-07

We would like to express our respect to all reviewers and AC. Thanks for your time and hard work. Based on your professional opinions, we have carefully replied all the high-value questions and supplemented the content accordingly.

---

> ### Author Response · Authors · 2024-08-14
> **Final reminder for LAST 3 Hrs of Discussion**
>
> Dear Reviewers,
>
> Sorry to bother you, but this is the last reminder from the authors. **We sincerely invite all reviewers to review and evaluate our responses**. We thank you for your hard work in the early stage and hope to get your short attention again.
>
> Since **there are only three hours left for discussion**, we attach great importance to this submission and would like to buy you a little time at the end.
>
> Warm regards,
>
> 9831 Authors

---

### Decision · Program_Chairs · 2024-09-25

**Decision:**

Accept (poster)

**Comment:**

This paper receives an overall rating of 4/5/5/5 (average: 4.75). After the rebuttal, two positive reviewers were generally satisfied with the rebuttal and chose to maintain their ratings, while two other reviewers did not respond. The AC read the paper and all discussions. Although the technical contribution is not significant, it was well-prepared and improved the accuracy of radar-data segmentation. Additionally, the AC thinks the main issues raised by the reviewers have been addressed. So, the AC recommends accepting this paper, and urges the authors to add the supplementary contents during the rebuttal to the final paper.